# SUBGRAPH GENERATION FOR GENERALIZING ON OUT-OF-DISTRIBUTION LINKS

## ABSTRACT

Graphs Neural Networks (GNNs) demonstrate high-performance on link prediction (LP) datasets, especially when the distribution of testing samples falls within the dataset's training distribution. However, GNNs suffer decreased performance when evaluated on samples from outside their training distribution. In addition, graph generative models (GGMs) show a pronounced ability to generate novel output graphs. Despite this, the application of GGMs remains largely limited to domain-specific tasks. To bridge this gap, we propose leveraging GGMs to produce synthetic samples which extrapolate between training and testing distributions. These synthetic samples are then used for fine-tuning GNNs to improve link prediction performance in out-of-distribution (OOD) scenarios. We introduce a theoretical perspective on this phenomena which is further verified empirically via increased performance across synthetic and real-world OOD settings. We conduct further analysis to investigate how inducing structural change within training samples improves OOD performance, indicating promising new developments in graph data augmentation on link structures.

## 1 INTRODUCTION

Graph Neural Networks (GNNs) demonstrate the ability to learn on graph data and have been used on a number of different downstream tasks that rely on understanding graph structure (Kipf & Welling, 2017). Link Prediction (LP)(Liben-Nowell & Kleinberg, 2003; Li et al., 2024), which attempts to predict unseen links in a graph, serves as one such example. For the task of LP, GNNs are used to learn node representations, which are then used to determine whether two nodes will form a link (Kipf & Welling, 2016). In recent years, advanced architectures have further enhanced state-of-the-art link prediction performance. To achieve this, the models often leverage structural features directly within their neural architecture, enabling the model's more effective understanding of link formation(Wang et al., 2023; Yun et al., 2021; Shomer et al., 2024).

However, recent studies indicate that GNNs struggle to generalize to out-of-distribution (OOD) samples. This can arise when the underlying dataset properties differ between training and testing (Gui et al., 2022). Additionally, the distribution shift in graph data is not well-aided by generalization techniques from other machine learning domains, such as CV and NLP (Li et al., 2022a; Gao et al., 2023). Therefore, the study of the OOD problem has flourished for graph- and node-classification (Ji et al., 2022; Koh et al., 2021). However, little direct attention has been paid to designing link prediction models which better withstand shifts in the underlying data distribution (Zhou et al., 2022; Bevilacqua et al., 2021). This is an issue, as recent work (Revolinsky et al., 2024) has shown that current link prediction models (even when augmented with OOD-generalization techniques) struggle to generalize to shifts in the underlying structural distribution. Given the success of out-of-distribution (OOD) generalization techniques in various graph-related tasks beyond link prediction (Arjovsky et al., 2019; Krueger et al., 2021; Wu et al., 2024; Wang et al., 2020), a question arises regarding the relatively limited success of these methods within the OOD link prediction problem. How can we improve out-of-distribution performance in link prediction?

Intrinsically, out-of-distribution problems are difficult to manage; the simplest solution is to retrain or tune the model on new samples within distribution of the testing set (Bai et al., 2023). Before retraining can occur, the samples must be acquired, or even detected that they fall out-of-distribution (Wu et al., 2023b;a). A promising example of this application occurs within both CV and NLP, where

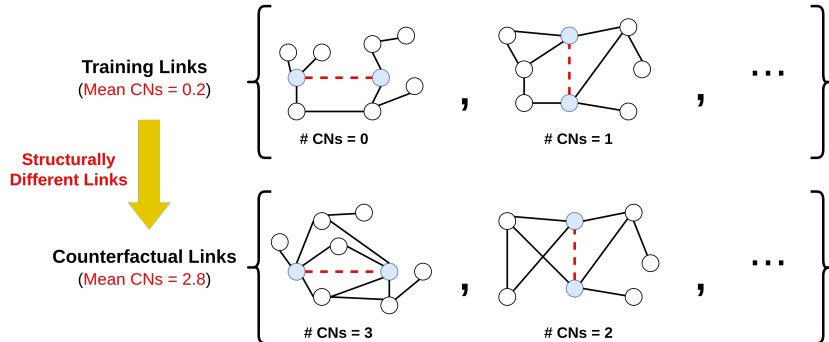

Figure 1: Example of counterfactual links that differ in terms of their structural properties such as Common Neighbors (CNs). In this example, the average training link typically contains very few CNs (0.2), thus we may want to generate counterfactuals with more CNs (2.8).

the training data is augmented with **counterfactual samples**. Such counterfactual samples have been shown to be helpful for OOD tasks by improving the diversity of the training data problem (Sun et al., 2022). This uplift is possible because counterfactual samples operate under the same causal rules as the original samples, even if the counterfactual sample was not originally contained within the training dataset (Ma et al., 2022). An example of how this may work for link prediction is shown in Figure 1, where the counterfactual links are meant to be structurally different from the training samples. As shown, the training samples have none or few common neighbors (i.e., shared 1-hop neighbors), the counterfactual samples have multiple. The counterfactual links thus demonstrate an *alternative reason* for why some links may form. Within link prediction, counterfactuals have demonstrated the ability to enhance baseline model performance (Zhao et al., 2022). However, these methods are often reliant on expensive pre-processing to generate counterfactuals, also requiring prior knowledge of the dataset's distribution shift, limiting real-world use (Zhao et al., 2022; Sun et al., 2022).

Thus, an important question is, *how can we learn to efficiently generate new but meaningfully different samples to improve LP generalization?* To address this issue, we apply graph generation as a data augmentation method to generate samples which are *counterfactual* to the training distribution. The underlying principle behind this approach is to determine if it is possible to augment our training distribution to increase generalization and potentially improve LP performance. In order to achieve this, we design a new framework called **FLEX** which leverages a generative graph model (GGM) co-trained with a GNN to produce subgraphs that are conditioned on a specific training link. The goal of the GGM is to take a single potential link (that is positive or negative) as input, and learn how to generate a new link that is counterfactual in structure to the input. To ensure that the GGM learns to generate counterfactual links, we maximize the Kullback-Leibler (KL) divergence with a quadratic penalty between posterior and prior sampling distributions to maximize structural diversity, but ensure we don't deviate too far from the original distribution. Furthermore, to avoid generating the entire adjacency for each new link, we instead propose to work with subgraphs, thus overcoming issues with efficiency.

Our contributions can be summarized as the following:

1. Overall, we introduce **FLEX**: a *simple yet effective* graph-generative framework that learns to generate counterfactual examples for improved link prediction performance.

2. We demonstrate the effect of structural shifts through targeted analysis on link prediction model performance.

3. We also conduct numerous experiments to show how FLEX can improve model generalization across multiple datasets and methods.

## 2 BACKGROUND AND RELATED WORK

We denote a graph as $\mathbf{G}(\mathbf{X}, \mathbf{A})$, abbreviated to $\mathbf{G}$, where $\mathbf{X} \in \mathbb{R}^{n \times d}$ represents the node features in real space with $n$ nodes and feature dimensions $d$. $\mathbf{A} \in \{0, 1\}^{n \times n}$ represents the adjacency matrix, within which nodes connect with one another to form edges, $e = (u, v)$. The $k$-hop subgraph of a node $v$ is denoted by $\mathbf{A}_v^{(k)}$. Consequently, the $k$-hop subgraph enclosed around an edge $e$ is defined as $\mathbf{A}_e^{(k)} = \mathbf{A}_u^{(k)} \cup \mathbf{A}_v^{(k)}$.

**Link Prediction**: Graph Neural Networks (GNNs) (Kipf & Welling, 2017) are a common tool for modeling link prediction. GNNs learn representations relevant to graph structure as embeddings, $\mathbf{H} = \mathrm{GNN}(\mathbf{X}, \mathbf{A})$ which are then passed to link predictors to estimate whether a link will form or not. However, several studies (Zhang et al., 2021; Srinivasan & Ribeiro, 2019) have shown that standard GNNs are not enough for link prediction, as the models ignore the pairwise information between two nodes. To account for this, recent methods either inject or augment pairwise information within GNNs to elevate their link prediction capabilities. We include more discussion link-prediction models within Appendix A.

**Graph Generative Models**: We treat graph generation as output of a scoring function $s : \mathbb{R}^d \times \mathbb{R}^d \to \mathbb{R}$ to quantify similarity between node embeddings, which is often defined as an inner product: $s(u, v) = \mathbf{H}_u^\top \mathbf{H}_v$ and further calculated as edge-probabilities, $P((u, v) \in E \mid \mathbf{H}_u, \mathbf{H}_v) = \sigma(s(i, j))$, where $\sigma(\cdot)$ is the sigmoid function. Whereas, we focus on the capability of auto-encoders inferring from latent embeddings to re-produce an adjacency matrix (Kipf & Welling, 2016). More advanced graph generation models exist: such as auto-regressive, diffusion, normalizing-flow, and generative-adversarial networks (You et al., 2018; Vignac et al., 2022; Luo et al., 2021; Martinkus et al., 2022). However, these models often employ mechanisms which restrict their applications beyond graph generation. For example, discrete-denoising models generate a new adjacency matrix with discrete space edits, which can be computationally restrictive to re-train when generalizing on a variety of different graph structures (Kong et al., 2023).

**Methods for OOD**: Numerous methods, operating underneath the invariance learning principle, exist to improve the generalization performance of neural models (Arjovsky et al., 2019). These invariant methods divide training data into environmental subsets for conditioning models to variance between training subsets. However, these methods require careful considerations for effective performance improvement in OOD scenarios (Gulrajani & Lopez-Paz, 2020). Additionally, generalizing with these techniques is difficult for graph representation learning (Li et al., 2022b; Revolinsky et al., 2024). Therefore, architectures and techniques which target invariance principles within graph data are employed to improve GNN performance (Chen et al., 2023; Zhang et al., 2022). Recently, graph generation has been applied within OOD scenarios as well. For example, EERM is a technique which integrates graph generators to improve OOD performance on graphs. However, the generators can lead to scalability issues when considering the additional nodes necessary for link formation (Wu et al., 2022). GOLD leverages latent generative models to learn on OOD samples, yet it functions predominantly for OOD detection on graphs and not directly improving OOD generalization in link prediction (Wang et al., 2025). Lastly, CFLP (Zhao et al., 2022) considers extracting counterfactual links for enhancing link prediction. However, their proposed algorithm is (a) a non-parametric method that relies on the Louvain (Blondel et al., 2008) algorithm, (b) has been shown to be prohibitive to run. This paper's initial runtime investigations verify CFLP's difficulty scaling within Appendix F, Tables 5 and 6.

## 3 FLEX

In Section 1, we introduced the OOD problem for link prediction and how graph generation has potential to solve the problem. However, *is it possible to generate such counterfactual links?* Effectively, there are endless "meaningless" graphs with no relevant structure to a training dataset; a GNN tuned on these graphs is also likely to suffer decreased downstream model performance. Therefore, applying graph data augmentation to improve performance requires understanding of the structure within the graph dataset (Singh et al., 2021). It's thus desirable for a learnable framework which understands link formation but can also target relevant graph structure to improve OOD performance. To achieve this, we introduce **FLEX**, the **F**ramework for **L**earning to **EX**trapolate

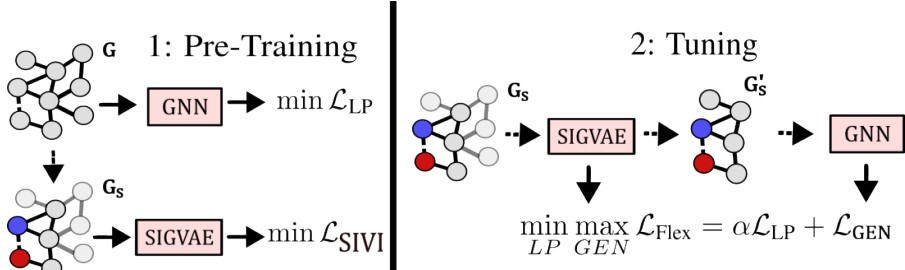

Figure 2: An illustration of the FLEX framework for a single dataset sample. **Step 1** involves pre-training both models separately to optimize their performance, like in real-world scenarios. **Step 2** involves adversarial co-training of the two models, where the GGM generates synthetic samples to tune the GNN.

Structures in Link Prediction. As a graph data augmentation framework, FLEX utilizes a variety of techniques to ensure: computability, scalability, and expressiveness.

Following these principles, FLEX then functions in two critical steps, as illustrated in Fig. 2. **First**, we pre-train a GNN on the dataset's full adjacency matrix by optimizing the predictive loss, $\mathcal{L}_{\mathrm{LP}}$. GNN pre-training simulates a real-world scenario, where we may only wish to improve a pre-existing model's ability to generalize on OOD samples (Gui et al., 2022; Krueger et al., 2021). A graph generative model (GGM) is then pre-trained separately to minimize generative loss, $\mathcal{L}_{\mathrm{SIVI}}$. The GGM is conditioned on each sample (i.e., link) via the labeling trick on the $k$-hop enclosed subgraph (Zhang et al., 2021). This ensures that we can generate a *new link* that is counterfactual to an *existing link*. **Second**, we apply both pre-trained models in a co-training framework, where the GGM produces synthetic dataset samples as input for fine-tuning the GNN. The GGM maximizes the distance between posterior and prior while the GNN attempts to minimize prediction loss; much like adversarial-conditioning in GANs and other auto-encoder frameworks (Goodfellow et al., 2020; Yang et al., 2019; Wang et al., 2025). As such, the GNN prediction loss functions to retain information from the original dataset distribution, further acting as counterfactual conditioning to improve OOD performance.

## 3.1 GENERAL MOTIVATION

The main objective of the FLEX framework is to generate graph samples which retain node feature properties while producing edge structures counterfactual to the original data. After which, the co-trained GNN is tuned on the synthetic counter-factual to improve performance. This is feasible with any type of well-trained graph generative model (e.g., auto-encoders (Kipf & Welling, 2016) or diffusion models (Vignac et al., 2022)). To explain what constitutes a relevant counterfactual for link prediction, we consider the following definitions.

**Definition 3.1** (Basic Counterfactual Entity). *Given a structural equation model $(M)$, consisting of two function sets $(Y, X)$. Let $M_x$ represent a modified version of $M$ where all possible $X = x$. When we infer $x$ from $Y$ with an input $u$, this represents the axiom: $Y_x(u) \triangleq \Delta Y_{M_x}(u)$ (Pearl, 2009).*

As such, Definition 3.1 represents the most basic example of a counterfactual, where $Y$ would properly denote the expected outcome $y$, had the function $X$ been $x$ for the given input $u$ (Pearl, 2013). In context of machine learning, this is further represented as a model learning a function which generalizes performance to testing data had training data formed differently.

To extend this for graph-structured data, specifically link prediction, we need an understanding of *what* our generated samples should be counterfactual to. Intuitively, we target higher-order link properties (Common Neighbors) which were previously unobserved within the training data. As shown in the next definition, an encoder $f_\theta(\cdot)$ that can extract expressive link features is therefore necessary for producing proper counterfactual links. If $f_\theta(\cdot)$ is not suitably expressive, our generative model will be unable to distinguish higher-order link structure and fail to generate counterfactuals relevant to the current model's training distribution.

**Definition 3.2** (Expressive Link Features). *Consider an edge sample $e = (u, v)$, and it's $k$-hop subgraph $\mathbf{A}_e^{(k)}$. We want to learn an encoder $f_\theta(\cdot)$ that can operate on $\mathbf{A}_e^{(k)}$ and learn to extract*

*structural features* $\mathbf{H}_e$ *that are specific to the link* $(u, v)$ *(e.g., link heuristics (Newman, 2001; Katz, 1953)). We assume that* $f_\theta(\cdot)$ *is expressive such that it can extract link-specific features. We then represent the probability distribution of the features extracted by the encoder to be* $\mathbb{P}_H(\mathbf{A}_e^{(k)}) = f_\theta(\mathbf{A}_e^{(k)})$.

**Definition 3.3** (Structural Link-Counterfactual). *For an edge sample* $e = (u, v)$, *a meaningfully different sample (counterfactual)* – $\tilde{\mathbf{A}}_e^{(k)}$ *exists where the link feature distribution estimated between the original subgraph and it's counterfactual are approximately non-equivalent,* $\mathbb{P}_H(\mathbf{A}_e^k) \not\approx \mathbb{P}_H(\tilde{\mathbf{A}}_e^{(k)})$.

A proper counterfactual sample should have different underlying link features from the original sample. As shown in Figure 1, we assume that we have an encoder which can extract common neighbors (CNs) (Newman, 2001). Given that the training samples have no or few CNs, the corresponding counterfactuals then contain a greater number of CNs. These new samples are thus *structurally-counterfactual*, in that they differ in higher-order structural features but retain the original node features.

**Corollary 3.3.1** (Feature-Conditional Equivalence). *Given the previous definition of counterfactual structure, the link features contained within* $k$*-hop subgraph* $\mathbf{A}_e^{(k)}$ *are not invariant in isolation as we must consider the node features. Therefore, in order for* $\tilde{\mathbf{A}}_e^{(k)}$ *to maintain a valid counterfactual structure, it must be conditioned on the node features* $\mathbf{X}_e^k$ *within the original subgraph. That is,* $\mathbb{P}_H(\mathbf{A}_e^{(k)} \mid \mathbf{X}_e^k) = f_\theta(\mathbf{A}_e^{(k)} \mid \mathbf{X}_e^k))$ *and* $\mathbb{P}_H(\tilde{\mathbf{A}}_e^{(k)} \mid \mathbf{X}_e^k) = f_\theta(\tilde{\mathbf{A}}_e^{(k)} \mid \mathbf{X}_e^k))$. *For convenience, we further write this as* $\mathbb{P}_H(\mathbf{G}_e^{(k)}) = f_\theta(\mathbf{G}_e^{(k)})$ *and* $\mathbb{P}_H(\tilde{\mathbf{G}}_e^{(k)}) = f_\theta(\tilde{\mathbf{G}}_e^{(k)})$.

Therefore, the link-counterfactual is dependent on the compatibility between $\tilde{\mathbf{A}}$ and $\mathbf{X}$. A failure to properly condition structure on $\mathbf{X}$ will not fulfill the definition for counterfactual structure since the newly-generated node features will introduce spurious correlations relative to original subgraph samples. So, the encoder $f_\theta(\cdot)$ must also consider the original node features as input. We further explain these principle within Appendix B.

Given these definitions, we can see generating proper counterfactual samples requires extracting link features conditional to node features. To do this, we learn a Generative Graph Model (GGM) which inputs both types of features to output a new sample with a different structural distribution. In order to do this, we must ensure three things: *(a) Scalability:* In order to ensure relevance to real-world problems, the GGMs must operate on large graphs. *(b) Expressiveness:* First, the extracted features for each link must be suitably expressive. Second, the GGM itself will need to effectively sample from complicated distributions to produce relevant graph structures. *(c) Counterfactual:* Generated structures must indicate a level of change which does not replicate the training distribution, but retains meaningful feature correlation. In the rest of this section, we outline our method for tackling these challenges. In consideration of space, we demonstrate the efficiency of our method within Appendix F.

### 3.2 SEMI-IMPLICIT VARIATION FOR OUT-OF-DISTRIBUTION GENERATION

Following principle (a.) from Section 3.1, the scalability of the practical implementation becomes a concern. Computational complexity of more refined GGMs can be restrictive, whereas less computationally-intensive generative models may result in low-quality generations (Simonovsky & Komodakis, 2018; Yan et al., 2024). To balance this, we employ semi-implicit variation (Yin & Zhou, 2018), for it's inherent scalability when implemented in an auto-encoder and it's expressiveness for modeling complex distributions.

Let the true data-generating distribution be $p(G)$, and assume it is modeled via a latent variable model with latent code $H$ and a semi-implicit posterior of the form:

$$q_\phi(H_e \mid \tilde{X}_e^{(k)}, \tilde{A}_e^{(k)}) = \int q_\phi(H_e \mid \psi) \, q_\phi(\psi \mid X_e^{(k)}, \tilde{A}_e^{(k)}) \, d\psi, \tag{1}$$

where $q_\phi(\psi \mid X, A)$ is a flexible (potentially implicit) distribution. Suppose the model is trained to maximize the semi-implicit evidence lower bound (ELBO) (Hasanzadeh et al., 2019):

$$\mathcal{L}_{\text{SIVI}} = \mathbb{E}_{\psi \sim q_\phi(\psi \mid X_e^{(k)}, A_e^{(k)})} \left[ \mathbb{E}_{H \sim q_\phi(H \mid \psi)} \left[ \log p(A_e^{(k)} \mid H_e) \right] - \text{KL}(q_\phi(H_e \mid \psi) \parallel p(H_e)) \right], \tag{2}$$

and assume $p(\mathbf{H}_e)$ is a broad prior (e.g., isotropic Gaussian) while $p(\mathbf{A_e} \mid \mathbf{H_e})$ defines a valid graph decoder. Then, given an auto-encoder with an expressive architecture capable of distinguishing the structure within samples drawn from $q_\phi$ and $p$, sampling from $\mathbf{H_e} \sim q_\phi(\mathbf{H}_e \mid \psi), \quad \psi \sim q_\phi(\psi)$ yields synthetic graphs $\tilde{\mathbf{G}}_\mathbf{e} = (\mathbf{X}_e, \tilde{\mathbf{A}}_\mathbf{e})$ whose features are derived from the original dataset distribution but reveal emergent out-of-distribution (OOD) structure with respect to the training data $\mathcal{D}_{\text{train}} \sim \mathbb{P}(\mathbf{G})$, provided that $q_\phi(\psi) \not\approx q_\phi(\psi \mid \mathcal{D}_{\text{train}})$. That is, the complete generative process follows:

$$\tilde{\mathbf{G}}_\mathbf{e} \sim p_\theta(\tilde{\mathbf{G}}_e \mid \mathbf{H}_e), \quad \mathbf{H}_e \sim q_\phi(\mathbf{H}_e \mid \psi), \; \psi \sim q_\phi(\psi), \tag{3}$$

Therefore, Eq. 3 defines a valid procedure for generating OOD graph samples. In scenarios where the sampled distribution is not a broad prior, this process then decomposes further to a standard variational generative process (Hasanzadeh et al., 2019; Kipf & Welling, 2016). We further develop our reasoning on link-counterfactual generative processes in Appendix B.

As a learnable mechanism, semi-implicit variance ($\psi$) often relies on inputting randomness into prior distributions; this randomness can then be treated as an adversarial noise, much like how OOD samples would appear to pre-trained GGMs. As such, an auto-encoder which effectively models semi-implicit variance of training distributions can generate complicated graph samples which mimic link-counterfactuals, fulfilling our expressiveness principle while maintaining the scalability of an auto-encoder (Hasanzadeh et al., 2019; Simonovsky & Komodakis, 2018). We show in Section 4.3 that the use of a semi-implicit GGM to a standard graph GGM is helpful for strong counterfactual generation.

## 3.3 LINK-SPECIFIC SUBGRAPH GENERATION

Semi-implicit variation assumes that a GGM can learn to generate $\tilde{\mathbf{G}}_\mathbf{e}$. However, as noted in Definition 3.2, to make this task relevant to link-prediction and continue fulfilling the expressiveness principle, we must first learn to extract *link-specific features*. That is, we want an encoder $f_\theta(G_e^{(k)})$ that can extract such features from the k-hop neighborhood of a link $e = (u, v)$. Only then will our GGM have the suitable amount of information to generate meaningful counterfactuals that differ in key link properties.

To achieve this, the encoder $f_\theta(\cdot)$ should be able to effectively encode the graph conditional on a specific link. The link-specific representations are then used by the GGM for generation. (Zhang et al., 2021) show that standard GNNs aren't expressive to links. To combat this, they introduce the labeling trick that ensures that a given GNN can learn to distinguish target links from other nodes within a graph sample. They demonstrate that the labeling trick can extract a number of different relevant structural features for a link (Zhang & Chen, 2018).

The labeling trick is defined as a function $\ell : \mathbf{A}^{(k)} \to \{0, 1\}$ where for a link $e = (u, v)$ the value for a sampled node $x$ is given by:

$$\ell(x) = \begin{cases} 1, & \text{if } x = u \; or \; x = v \\ 0, & \text{else} \end{cases} \tag{4}$$

This results in a labelled subgraph $L_e^{(k)}$ which is fed, along with the node features, to a GNN to produce the link-specific representations:

$$\mathbf{H}_e = \text{GNN}(L_e^{(k)}, X_e^{(k)}). \tag{5}$$

Given that all edges within a graph are viable link prediction targets, an effective zero-one labeling requires extracting the $k$-hop enclosed subgraphs conditioned on a target edge, $\mathbf{G}_e^{(k)}$. When these subgraphs are restricted to a smaller size, this reduces the direct computation required from the GGM to model subgraph distributions, ensuring FLEX's scalability principle (Zhang & Chen, 2018).

### 3.3.1 NODE-AWARE DECODER

Furthermore, to continue ensuring scalability and expressiveness. The decoder for FLEX's GGM is made aware of the independent number of nodes within subgraph samples for a given mini-batch along the block diagonal matrix, $A = \text{diag}(A_1, \ldots, A_K)$ with $A_i \in \mathbb{R}^{\mathcal{N}_i \times \mathcal{N}_i}$. This ensures that generated subgraphs retain the original number of input nodes and prevent message-passing along edges between distinct subgraph samples.

Within early experiments, as shown in Figure 13, generated subgraph samples suffered from the degree-bias phenomenon (Tang et al., 2020). Wherein, the backbone GNN learns on nodes with a higher number of edges at a much-greater frequency than low-degree nodes, prioritizing learning information from the high-degree nodes (Liu et al., 2023). Therefore, generated subgraph samples were always dense, regardless of the input graph's node-degree. We verify this phenomenon in Appendix M. To account for this, we apply an indicator function to FLEX-generated subgraphs which eliminates edges with lower probability than a threshold, $\gamma$:

$$\tilde{p}(u, v) = p(u, v) \cdot \mathbb{I}[p(u, v) \geq \gamma]. \tag{6}$$

This function only keeps those links with high probability, constraining the GGM to connect links which it is most confident in. As such, the indicator function prevents densely-connected graphs, especially for OOD scenarios where training on dense graphs may not be desirable for downstream performance. The value of the threshold $\gamma$ is treated as a hyperparameter. In Section 4.3, we show how the value of $\gamma$ impacts performance.

### 3.4 GENERATING COUNTERFACTUAL LINKS

As part of FLEX, all previous components work to produce meaningful subgraphs. However, it is still necessary for the GGM to learn how to produce subgraph samples which are structurally-dissimilar from training, while retaining relevance to the node features within the training distribution.

As discussed in Definition 3.3, to ensure generated samples are link-counterfactual we can input links structural feature distribution. That is, for an input training sample $e = (u, v)$ and it's counterfactual, we want that $\mathbb{P}_H(\mathbf{A}_e^k) \not\approx \mathbb{P}_H(\tilde{\mathbf{A}}_e^{(k)})$ where $\tilde{\mathbf{A}}_e^{(k)} = p_\theta(\tilde{\mathbf{G}}_\mathbf{e} \mid \mathbf{H}_e)$. That is, we need to optimize the GGM to maximize the difference in input and generated samples; $\max \mathcal{L}_{\text{GEN}}$ where $\mathcal{L}_{\text{GEN}}$ is defined as in Eq. 7.

However, blindly maximizing the generative loss will result in generated subgraphs which are structurally-incoherent to our training samples and therefore our baseline model. In reality, we nudge the generated sample distribution to modestly differ in key structural features. We ensure this in two ways. First, we apply a quadratic penalty to the generative loss $\mathcal{L}_{\text{GEN}}$. The penalty is centered around a target value, $\tau$. This penalty restricts any shifts to the posterior distribution. In effect, generated graphs will only deviate slowly from the prior distribution and prevent the samples from devolving into noise. This is given by the following,

$$\mathcal{L}_{\text{GEN}} = - \left( \mathcal{L}_{\text{SIVI}} - \text{KL} \left( \mathbb{E}_{\psi \sim q_\phi(\psi \mid X_e, A_e)} \left[ q(H_e \mid \psi) \right] \, \big\| \, p(H_e) \right) - \tau \right)^2. \tag{7}$$

Second, we also attempt to correctly classify the link based on it's original label. That is, we want to predict the existence of the original link based on the newly generated sample. This serves as a means for inducing learnable counterfactual treatment within the GGM. If the generative model deviates too far from the training distribution or considers useless structural features, the GNN will be unable to cope, thus resulting in poor classification performance. It therefore allows for a "check" on the generation quality, limiting the potential for incoherent generation.

The final optimization goal of FLEX is given by the following, $\mathcal{L}_{\text{LP}}$ denotes the classification loss (BCE):

$$\min_{LP} \max_{GEN} \mathcal{L}_{\text{Flex}} = \alpha \mathcal{L}_{\text{LP}} + \mathcal{L}_{\text{GEN}} \tag{8}$$

$\alpha$ represents the weight assigned to the counterfactual predictions produced by the GNN tuned within the FLEX framework. Since the co-trained GNN is tuned on synthetic samples, the minimization of $\mathcal{L}_{\text{LP}}$ ensures that the GNN retains it's ability to predict on positive and negative samples while also conditioning the maximization of $\mathcal{L}_{\text{GEN}}$. In tandem, the two function in an adversarial co-optimization to predict on samples with increasingly different structures (Pan et al., 2018; Wang et al., 2025).

We further illustrate the overall framework in Figure 2. In the first stage both the GNN and GGM are trained separately. Then in the second stage, the components are co-trained via the objective defined in Equation 8. Both procedures are described further in Algorithm 1. In the next section, we test FLEX, showing it's ability to improve OOD performance for link prediction.

# 4 EXPERIMENTS

We now evaluate FLEX to answer the following research questions. **RQ1:** Does FLEX contribute to better link prediction performance in OOD scenarios? **RQ2:** How might separate components of the FLEX framework improve OOD performance? **RQ3:** How sensitive is FLEX to different hyperparameter settings? **RQ4:** Does FLEX learn to generate link-counterfactual samples?

## 4.1 SETUP

Our benchmarking experiments apply two different GNN backbones, Graph Convolutional Network (GCN) and Neural Common Neighbor (NCN) (Kipf & Welling, 2017; Wang et al., 2023). We then compare against the following generalization methods: CORAL, DANN, GroupDRO, VREx, IRM (Sun & Saenko, 2016; Ganin et al., 2016; Sagawa et al., 2019; Krueger et al., 2021; Arjovsky et al., 2019). Detailed hyperparameter settings are included within Appendix G. For datasets, we consider the synthetic datasets generated via the protocol designed by LPShift (Revolinsky et al., 2024). Please see Appendix H for more details. As a means of testing performance under distribution shift, we test on the original ogbl-collab split (Hu et al., 2020) and domain-transfer between Amazon Photos and Computer (Shchur et al., 2018). Lastly, all synthetic datasets are evaluated using Hits@20, while ogbl-collab is evaluated with Hits@50 and domain-transfer with AUC.

## 4.2 RQ1: FLEX PERFORMANCE

As shown in Table 3, FLEX improves the performance in 28 out of 29 data scenarios when applied to GCN, and for all tested scenarios when applied to NCN. This leads to an average relative increase of **4.41% to GCN and 9.56% to NCN**. On the other hand, other baselines either perform worse or on-par with GCN. This indicates that FLEX generates subgraphs which improve model generalization under distribution shift.

Table 1: Hits@20 results for real-world and LPShift datasets, AUC results for domain-transfer datasets. LPShift dataset splits are marked "Forward" and "Backward", "Forward" meaning more higher-order structure within testing versus training, and vice versa for "Backward". CN = Common Neighbors, PA = Preferential-Attachment, SP = Shortest-Path. LPShift results are averaged across five datasets (Collab, PubMed, Cora, CiteSeer, PPA).

| Datasets | | | | | Methods | | | | |
| Type | Name | Metric | Avg. OOD | VGAE | CFLP | GCN | GCN+FLEX | NCN | NCN+FLEX |
|---|---|---|---|---|---|---|---|---|---|
| Forward | CN | Hits@20 | 51.07 ± 1.88 | 50.71 ± 1.06 | 53.70 ± 1.90 | 53.61 ± 1.13 | 54.43 ± 0.33 | 50.47 ± 2.24 | 52.55 ± 0.27 |
| | PA | Hits@20 | 62.99 ± 3.09 | 63.36 ± 2.01 | 67.61 ± 3.71 | 67.47 ± 2.66 | 68.86 ± 1.87 | 68.27 ± 0.87 | 68.97 ± 0.19 |
| | SP | Hits@20 | 41.70 ± 2.48 | 46.89 ± 1.60 | 35.64 ± 2.51 | 44.27 ± 2.36 | 46.56 ± 1.29 | 46.63 ± 2.00 | 52.46 ± 6.10 |
| Backward | CN | Hits@20 | 27.44 ± 2.30 | 26.29 ± 2.03 | 27.46 ± 0.99 | 29.69 ± 1.71 | 31.57 ± 0.43 | 22.06 ± 1.66 | 24.33 ± 1.33 |
| | PA | Hits@20 | 37.49 ± 2.45 | 31.97 ± 1.30 | 38.92 ± 1.86 | 44.52 ± 1.66 | 43.82 ± 1.53 | 38.19 ± 4.05 | 41.30 ± 0.11 |
| | SP | Hits@20 | 23.86 ± 2.79 | 26.28 ± 2.75 | 23.07 ± 1.89 | 24.96 ± 2.70 | 27.22 ± 0.76 | 22.61 ± 2.41 | 28.09 ± 0.86 |
| Real | Collab | Hits@50 | 47.98 ± 1.02 | 50.71 ± 0.21 | OOM | 50.40 ± 1.01 | 52.42 ± 0.08 | 64.83 ± 0.18 | 64.99 ± 0.32 |
| | P → C | AUC | 85.80 ± 3.52 | 88.94 ± 1.06 | OOT | 87.48 ± 2.73 | 91.16 ± 1.24 | – | – |
| | C → P | AUC | 82.58 ± 4.61 | 86.44 ± 3.15 | OOT | 83.87 ± 5.08 | 91.36 ± 0.05 | – | – |
| | Avg (Δ%) | – | | -7.44 | -7.09 | -0.19 | – | **+4.41** | – | **+9.56** |

## 4.3 RQ2: FRAMEWORK ABLATION

In order to determine which components of FLEX function to improve performance, we ablate across singular mechanisms which are directly involved with the FLEX-tuning process for the co-trained GNN. This includes the use of (a) semi-implicit variation, (b) an expressive link encoder (SEAL), (c) the LP loss $\mathcal{L}_{LP}$ described in Eq. 8. As shown in Table 2, ablating each component leads to a consistent decrease on four different datasets, thus validating the importance of each component.

Table 2: Ablation across the LPShift "Backwards" CN Splits.

| Dataset | | Models | | |
|---|---|---|---|---|
| | **FLEX** | w/o SEAL | w/o LP Loss | w/o SIGVAE |
| Cora | **44.87** ± 0.32 | 34.62 ± 0.49 | 39.15 ± 1.31 | 33.90 ± 0.35 |
| CiteSeer | **51.98** ± 0.03 | 41.63 ± 0.37 | 51.83 ± 0.24 | 41.58 ± 0.01 |
| PubMed | **29.31** ± 0.12 | 28.07 ± 0.12 | 28.66 ± 0.57 | 27.95 ± 0.08 |
| Collab | **25.24** ± 0.01 | 24.76 ± 0.03 | 24.78 ± 0.69 | 24.80 ± 0.69 |

### 4.4 RQ3: HYPERPARAMETER SENSITIVITY

In order to gauge the impact the that Eq. 7 has on downstream performance for FLEX, we conduct a study which measures the difference in performance across the indicator function's target $\gamma = \{0.0, 0.25, 0.5, 0.75, 0.9, 0.9999\}$. As shown in Figure 3, we see that the "Backward" split experiences gradually increasing performance up to a value of $0.9$ while the "Forward" split performance sharply decreases at a threshold value of $0.9999$. Given that indicator threshold values directly affect edge-probabilities, these results demonstrate that sparser generated graphs are useful for the "Backward" split to a point. Whereas little seems to affect a change in the "Forward" split performance until the graph grows too sparse at $0.9999$. We also include the effect of the learning rate in Figure 9.

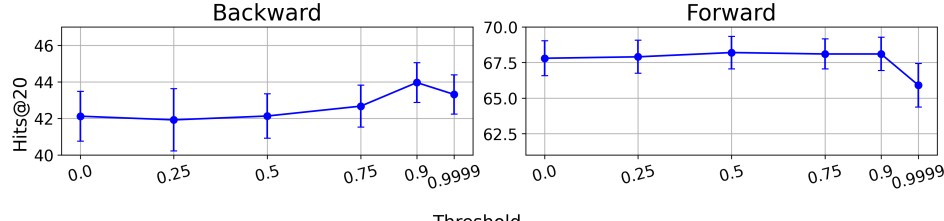

Figure 3: Performance of FLEX on the "Backwards CN" CiteSeer dataset across thresholds.

### 4.5 RQ4: OOD STRUCTURAL ALIGNMENT

To further verify the effect that FLEX has on graph structure and whether it generates samples with counterfactual link-structure, we directly measure the distribution of Common Neighbors within the original training and validation distribution versus FLEX-generated subgraphs. As shown in Figure 4, the "Flex - Generated" sample distribution closely matches the distribution of validation samples for the "Backward" subplot, with none of the FLEX samples exceeding a difference of 0.17 CNs. This is a 3-10x improved alignment versus the original training distribution. Within the "Forward" split, FLEX samples are verifiably denser than the 0 CNs present in training. Despite this, the threshold function still manages to ensure that FLEX samples never exceed a CN threshold of 1. This indicates that FLEX is **successfully targeting structure to produce graphs which are link-counterfactual to the training distribution** and help improve performance. A core consideration is FLEX's ability to do this without requiring access to validation or testing samples. We include more results on how FLEX affects node-degree and clustering coefficient within Appendix D.

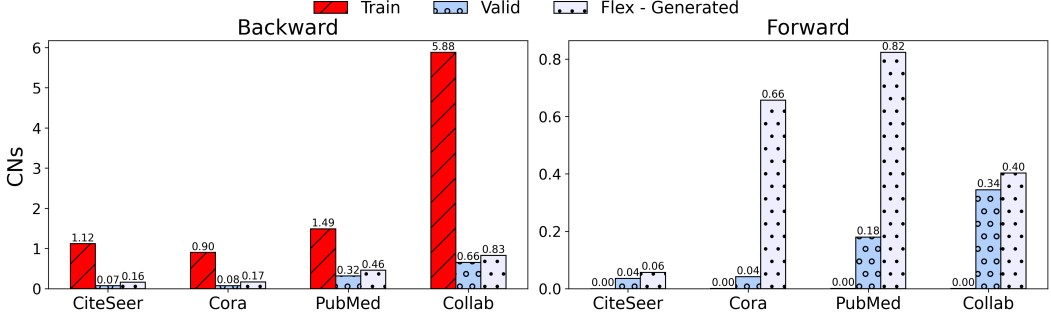

Figure 4: The distribution of Common Neighbors (CNs) scores across different dataset splits for the Backward and Forward CN LPshift splits.

## 5 CONCLUSION

Within this work, we formalize a theory for generating link-counterfactuals. To test this theory, we introduce FLEX, a simple generative framework which targets link-structures within input samples to

produce link-counterfactuals which improve downstream performance. Further experimentation indicates FLEX's ability to model OOD structures without access to validation and testing distributions. Additionally, tuning within the FLEX framework improves performance under realistic and synthetic distribution shifts, even where traditional generalization methods often decrease performance. This work opens considerations on the application of graph generation with distribution shifted scenarios, potentially opening a path to further development of counterfactuals within graph representations.

# 6 LLM USAGE DISCLOSURE

We use LLMs solely as writing-assist and coding-assist tools to polish the manuscript and debug broken functionality within this research's code. LLMs were used to fix broken formatting within LaTeX and resolve persistent dataloading issues. All research ideas, methodology, experiments, theoretical analyses, and initial drafts were conceived and written by the authors.

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

## A   RELATED WORKS - CONTINUED

There are numerous models and methods to improve the link-prediction capabilities of GNNs. First of which include SEAL (Zhang & Chen, 2018) and NBFNet (Zhu et al., 2021), which consider message passing schemes that are conditional on a given link. To improve efficiency, other methods don't modify the message passing process, instead opting to include some link-specific information when scoring a prospective link. BUDDY applies a unique version of the labeling trick to subgraphs for generalizing on structural features (Chamberlain et al., 2022). NCN/NCNC (Wang et al., 2023) and Neo-GNN (Yun et al., 2021) both elevate traditional link heuristics via neural operators to better understand link formation. Lastly, (Shomer et al., 2024) proposes a more general scheme for estimating the pairwise information between nodes that adaptively learns how two nodes relate. A core component of these models is their increased reliance on the substructures contained within the graph datasets, which improves the model's expressivity but can affect prediction performance in OOD scenarios (Mao et al., 2024).

## B   SET-THEORY PERSPECTIVE

Within the following section, we detail how treating the space of training and test samples within the domain of their node features can feasibly lead to scenarios where a GGM will produce link-counterfactual samples which extend the scope of the training distribution with the testing distribution.

**Definition B.1** (Node-feature domain and link distributions). *Let $\mathbf{X} \subseteq \mathbb{R}^d$ be the node-feature space. A link is an element of $\mathbf{X} \times \mathbf{X}$. Let $P_{\text{train}}$ and $P_{\text{test}}$ be probability measures on $\mathbf{X} \times \mathbf{X}$ with supports*

$$T := \text{supp}(P_{\text{train}}), \qquad U := \text{supp}(P_{\text{test}}).$$

*Remark* 1. In Figure 5, $T$ (blue) and $U$ (red) are subsets of the same domain; their overlap $T \cap U$ is visualized by triangle hatching.

**Assumption 1** (Link-counterfactual conditioning mechanism). *There exists a counterfactual mechanism $\mathcal{C}$ that, given samples from $P_{\text{train}}$ and link structure, produces link-counterfactuals samples in a set $S \subseteq \mathbf{X} \times \mathbf{X}$. We assume $T \subseteq T' := \overline{T \cup S}$ (closure taken in $\mathbf{X} \times \mathbf{X}$). Operationally, $\mathcal{C}$ may be implemented by counterfactual structural perturbations parametrized by ELBO-guided sampling under learned generative constraints. In Figure 6, $S$ is indicated by square hatching surrounding $T$ (yellow annulus).*

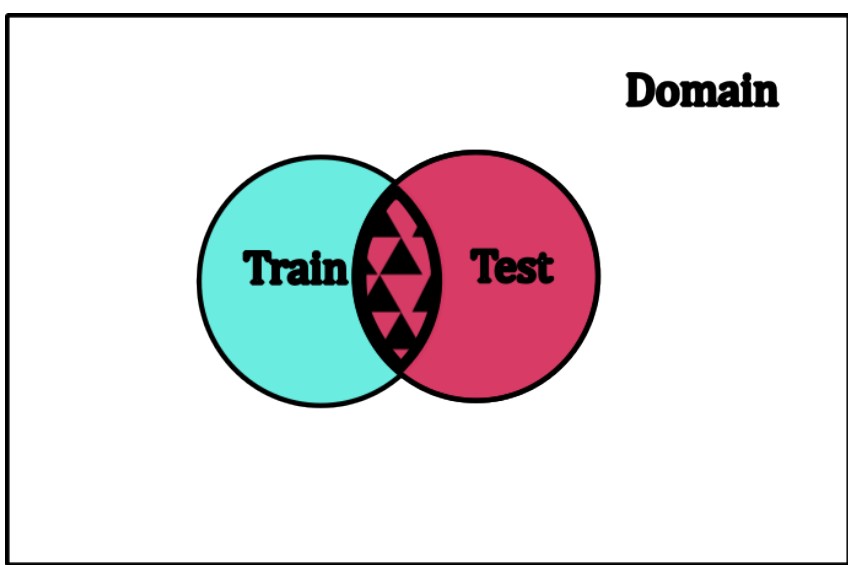

Figure 5: The domain space depicting $T$ (Train) and $U$ (Test) with triangle hatching for $T \cap U$.

**Definition B.2** (Overlap measure). *Let $\mu$ be the ambient Lebesgue measure on $\mathbb{R}^{2d}$ (or any measure absolutely continuous with respect to both $P_{\text{train}}$ and $P_{\text{test}}$). Define the overlap sizes*

$$\Omega(T, U) := \mu(T \cap U), \qquad \Omega(T', U) := \mu(T' \cap U).$$

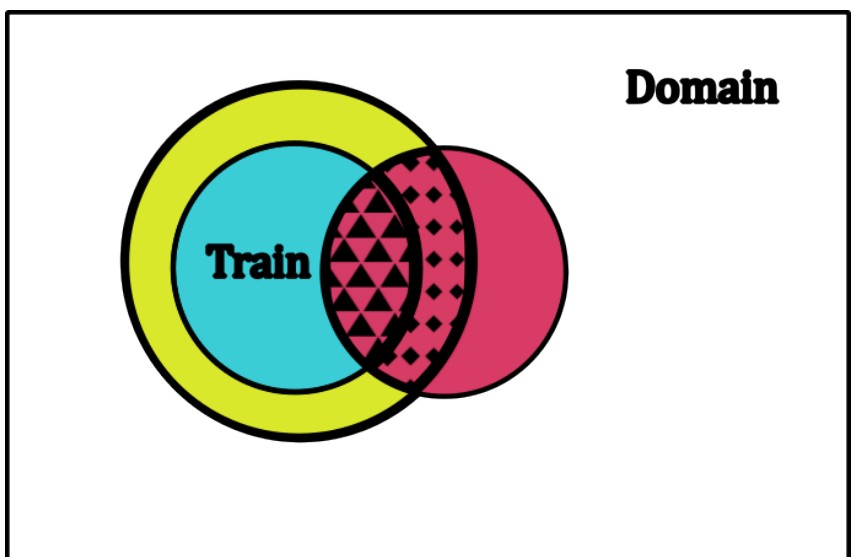

Figure 6: The domain space extended from Figure 5. The larger (yellow) set encapsulating $T$ demonstrates the expansion to $T'$ via $S$ (square hatching), increasing the overlap with $U$ as guaranteed by Theorem 1

**Theorem 1** (Coverage expansion via structural conditioning). *Under the Structural Conditioning Assumption, if the conditiond set intersects the OOD region with positive measure,*

$$\mu\big(S \cap (U \setminus T)\big) \; > \; 0,$$

*then the training–test overlap strictly increases:*

$$\Omega(T', U) \; > \; \Omega(T, U).$$

*Proof.* By definition $T' = \overline{T \cup S}$ and $T \subseteq T'$. Hence

$$T' \cap U \; = \; \overline{(T \cup S)} \cap U \; \supseteq \; (T \cup S) \cap U \; = \; (T \cap U) \cup (S \cap U).$$

Taking $\mu$ and using subadditivity with the union decomposition,

$$\mu(T' \cap U) \; \geq \; \mu(T \cap U) + \mu\big(S \cap U \setminus (T \cap U)\big).$$

Note that $S \cap U \setminus (T \cap U) = S \cap (U \setminus T)$. By our hypothesis $\mu\big(S \cap (U \setminus T)\big) > 0$, therefore

$$\mu(T' \cap U) \; > \; \mu(T \cap U).$$

Equivalently, $\Omega(T', U) > \Omega(T, U)$, proving the claim. $\qquad\square$

**Corollary B.2.1** (Bayesian consequence for generalization). *Assume a model class with likelihood $p_\theta$ is trained only on $P_{\text{train}}$ (or its empirical sample) to form a posterior $p(\theta \mid \mathcal{D}_{\text{train}})$. If Theorem 1 holds, then evaluating on $P_{\text{test}}$ after augmenting training with link-counterfactual samples from $S$ reduces the* measure *of purely OOD inputs $U \setminus T'$ compared to $U \setminus T$. Consequently, any risk functional that is nonnegative and integrates over test support (e.g., expected loss) can only benefit from the reduction of the OOD region, all else equal.*

**Definition B.3** (Structural hull of training support). *Let $\Pi$ be a family of structure-preserving perturbations (e.g., counterfactual edits that obey graph constraints such as Common Neighbors). Each $\pi \in \Pi$ induces a measurable map $\Phi_\pi : \mathbf{X} \times \mathbf{X} \to \mathbf{X} \times \mathbf{X}$. The* structural hull *of $T = \text{supp}(P_{\text{train}})$ is*

$$\mathsf{Hull}_\Pi(T) \; := \; \overline{\{\Phi_\pi(x) \, : \, x \in T, \, \pi \in \Pi\}} \; \subseteq \; \mathbf{X} \times \mathbf{X}.$$

**Assumption 2** (Encoding Continuous Embeddings). *Given our encoding scheme $(\Pi : G \to \mathbf{H})$, the sets $(T, U, S)$ are mapped into continuous representation space $(H \subseteq \mathbb{R}^{2d})$ (i.e, a continuous latent embedding space). Therefore, enabling the ability to affect coverage given the ambient Lebesque measure, $\mu$. We treat $\mathbf{X}$ and $\mathbf{H}$ interchangeably.*

**Assumption 3** (ELBO-trained generator with structural constraints). *Let $p_\theta(x \mid z)$ be a decoder likelihood on $\mathbf{X} \times \mathbf{X}$ with latent prior $p(z)$, and let $q_\phi(z \mid x)$ be a variational encoder. Training maximizes the ELBO over $\mathcal{D}_{\text{train}}$, possibly augmented with structure-preserving perturbations $\Pi$:*

$$\mathcal{L}_{\text{ELBO}}(\theta, \phi) = \mathbb{E}_{x \sim P_{\text{train}}} \Big[ \mathbb{E}_{z \sim q_\phi(\cdot \mid x)} \big[ \log p_\theta(x \mid z) \big] - \text{KL}\big(q_\phi(z \mid x) \,\|\, p(z)\big) \Big]$$

$$\textit{subject to } x \in \text{Hull}_\Pi(T).$$

*Sampling link-counterfactual points is implemented by: draw $x \sim P_{\text{train}}$, choose $\pi \in \Pi$, form $\tilde{x} = \Phi_\pi(x) \in \text{Hull}_\Pi(T)$, then sample $z \sim q_\phi(\cdot \mid \tilde{x})$ and emit $\hat{x} \sim p_\theta(\cdot \mid z)$. Let $S$ be the set of realizations of $\hat{x}$ with non-negligible likelihood under the trained $(\theta, \phi)$.*

**Assumption 4** (Support-positivity and absolute continuity). *(i) $p_\theta(x \mid z) > 0$ for all $x$ in an open neighborhood of $\text{Hull}_\Pi(T)$ for $q_\phi$-a.e. $z$ (decoder has positive density on a structural neighborhood). (ii) $P_{\text{test}}$ is absolutely continuous with respect to $\mu$ (the ambient Lebesgue measure on $\mathbb{R}^{2d}$). (iii) There exists a set $W \subseteq \text{Hull}_\Pi(T) \cap U$ with $\mu(W) > 0$ such that $\inf_{x \in W} \mathbb{E}_{z \sim q_\phi(\cdot \mid x)}[p_\theta(x \mid z)] > 0$ (posterior predictive places nonzero mass on a test-overlapping region of the hull).*

**Lemma 1** (ELBO-guided structural conditioning yields positive OOD coverage). *Under the above assumptions, the structurally-conditioned sample set $S$ satisfies*

$$\mu\big(S \cap (U \setminus T)\big) > 0.$$

*Consequently, the hypothesis of Theorem 1 holds, and the training–test overlap strictly increases: $\Omega(T', U) > \Omega(T, U)$.*

The question still remains, how do we extend these Set-theoretic principles into the discrete domain for generating link-counterfactuals which can improve OOD performance?

*Proof sketch.* By construction, realizations $\hat{x}$ concentrate where the joint $q_\phi(z \mid x) p_\theta(\hat{x} \mid z)$ is large with $x \in \text{Hull}_\Pi(T)$. Assumption 3(i) implies that for any measurable $\mathbf{A} \subset \text{Hull}_\Pi(T)$ with $\mu(\mathbf{A}) > 0$, the decoder assigns strictly positive probability to neighborhoods within $\mathbf{A}$. By 3(iii), there exists $W \subseteq \text{Hull}_\Pi(T) \cap U$ with $\mu(W) > 0$ on which the posterior predictive is uniformly positive, so samples land in $W$ with nonzero probability. Since $W \subseteq U$ and, by defintion of OOD, $U \setminus T$ has positive $\mu$-measure in typical OOD scenarios (Figure 6), we obtain $\mu(S \cap (U \setminus T)) > 0$. Therefore the sufficient condition of Theorem 1 is met. $\qquad\square$

*Remark* 2 (Operational Takeaway #1). If your generator is trained with ELBO while respecting structural perturbations $\Pi$, and the decoder retains positive density on a neighborhood of the structural hull, then *sampling through the encoder–decoder pipeline from structurally-perturbed points* produces a set $S$ that (with positive measure) reaches into the OOD region $U \setminus T$, thus enlarging coverage and improving test overlap.

*Remark* 3 (Operational Takeaway #2). Genuinely ensuring that learned parametrizations of structural perturbations $\Pi$ always increase coverage to OOD regions/datasets is difficult in practice, since $\mu$-measure for all possible OOD samples are inaccessible or have limited accessibility from the training distribution. Careful considerations about dataset balance must be considered (i.e. smaller structures in training samples have less to infer for structure in larger testing samples)

We further formalize the intuition that augmenting training support broadens test risk to improve coverage within OOD scenarios.

**Proposition B.1** (Coverage-based OOD risk bound). *Let $\ell \in [0, 1]$. For any predictor $f$,*

$$R_{\text{test}}(f) \leq R_{\text{train}'}(f) + \delta',$$

*where $\delta' := P_{\text{test}}(N')$ and $N' = \text{supp}(P_{\text{test}}) \setminus \text{supp}(P_{\text{train}'})$.*

*Proof sketch.* Decompose the test risk over the covered and uncovered regions:

$$R_{\text{test}}(f) = \int_{C'} \ell(f, x) \, dP_{\text{test}}(x) + \int_{N'} \ell(f, x) \, dP_{\text{test}}(x).$$

On $C'$, the density of $P_{\text{test}}$ is supported inside $\text{supp}(P_{\text{train}'})$, so we can rewrite

$$\int_{C'} \ell(f,x) \, dP_{\text{test}}(x) \; \leq \; R_{\text{train}'}(f)$$

up to standard estimation error terms (handled separately by classical generalization bounds). On $N'$, we only know that $\ell \leq 1$, hence

$$\int_{N'} \ell(f,x) \, dP_{\text{test}}(x) \; \leq \; \int_{N'} 1 \, dP_{\text{test}}(x) \; = \; P_{\text{test}}(N') \; = \; \delta'.$$

Combining the two inequalities yields $R_{\text{test}}(f) \leq R_{\text{train}'}(f) + \delta'$. $\qquad\qquad\qquad\square$

**Corollary B.3.1** (FLEX shrinks the uncovered test mass). *Assume the conditions of Lemma 1 and Theorem 1, and suppose $P_{\text{test}}$ is absolutely continuous w.r.t. $\mu$ on $U$. Let*

$$\delta := P_{\text{test}}\big(U \setminus T\big), \quad \delta' := P_{\text{test}}\big(U \setminus T'\big).$$

*If $\mu\big(S \cap (U \setminus T)\big) > 0$, then*

$$\delta' < \delta.$$

*Consequently, for any predictor $f$,*

$$R_{\text{test}}(f) \; \leq \; R_{\text{train}'}(f) + \delta' \; < \; R_{\text{train}'}(f) + \delta.$$

*Remark* 4 (KL-regularization as a surrogate for coverage expansion). Our FLEX objective maximizes a KL divergence $\text{KL}(P_* \,\|\, P_{train})$ constrained to the structural hull $\text{Hull}_\Pi(T)$. Under mild regularity conditions, any nontrivial increase in this KL divergence implies that the counterfactual distribution $P_*$ assigns positive probability to regions of $U \setminus T$ with $\mu(\cdot) > 0$, thus increasing $\mu(S \cap (U \setminus T))$ for the resulting sample set $S$. The coverage expansion guaranteed by Lemma 1 then translates, via Proposition B.1, into a strictly tighter upper bound on our test risk.

## C  RAW RESULTS TABLES

Table 3: Results for the LPShift Datasets by direction (*forward* or *backwards*) and type (*CN, SP* or *PA*). OOT = Out-of-Time, OOM = Out-of-Memory. Ordered from bottom up: Collab, PubMed, Cora, CiteSeer, PPA. *Note:* PPA for PA and SP is missing due to taking >24h. Results for the original ogbl-collab (Hu et al., 2020) are included as *real*. Cross-Domain transfer dataset performance is measured after one-shot tuning on top of an already-tuned baseline. We highlight in blue when FLEX increases over the base model and red otherwise.

| Dataset | | CORAL | DANN | GroupDRO | VREx | IRM | VGAE | CFLP | GCN | GCN+FLEX | NCN | NCN+FLEX | EERM | HL-GNN | HL-GNN+FLEX | GAT | GAT+FLEX | GIN | GIN+FLEX |
|---|---|---|---|---|---|---|---|---|---|---|---|---|---|---|---|---|---|---|---|
| Forward | CN | 30.93 ± 0.24 | 30.86 ± 0.32 | 27.83 ± 1.76 | 30.93 ± 0.24 | 25.78 ± 2.04 | 21.60 ± 0.20 | OOM | 31.92 ± 0.25 | 32.87 ± 0.23 | 1.62 ± 5.04 | 3.95 ± 0.75 | OOM | 1.46 ± 2.19 | 2.57 ± 1.27 | OOM | – | 30.92 ± 0.58 | 31.73 ± 0.30 |
| | | 67.75 ± 2.49 | 68.11 ± 3.04 | 65.27 ± 3.50 | 66.54 ± 2.42 | 66.67 ± 1.50 | 71.32 ± 1.99 | 67.63 ± 2.51 | 67.18 ± 2.43 | 68.24 ± 1.30 | 75.83 ± 4.42 | 79.34 ± 0.1 | 55.93 ± 1.49 | 67.71 ± 2.52 | 68.23 ± 1.27 | 47.16 ± 4.63 | 49.03 ± 3.72 | 56.15 ± 2.89 | 58.07 ± 1.03 |
| | | 57.45 ± 1.70 | 57.54 ± 2.80 | 38.21 ± 5.63 | 53.15 ± 3.58 | 55.30 ± 2.54 | 54.74 ± 2.24 | 56.64 ± 1.29 | 56.22 ± 1.31 | 57.78 ± 0.08 | 75.91 ± 1.50 | 79.34 ± 0.10 | 66.89 ± 2.78 | 50.81 ± 4.91 | 55.19 ± 0.34 | 31.10 ± 3.26 | 52.47 ± 2.83 | 39.25 ± 4.78 | 41.05 ± 2.83 |
| | | 71.32 ± 0.32 | 71.62 ± 0.42 | 57.25 ± 1.95 | 71.60 ± 0.66 | 68.18 ± 0.48 | 65.18 ± 0.56 | OOT | 69.60 ± 0.45 | 70.04 ± 0.01 | 96.63 ± 0.24 | 96.72 ± 0.32 | OOM | 57.39 ± 2.14 | 58.01 ± 1.83 | 48.87 ± 1.06 | 49.62 ± 1.04 | 55.93 ± 16.06 | 59.19 ± 12.90 |
| | | 42.60 ± 1.61 | 43.60 ± 1.21 | 22.71 ± 4.03 | 43.61 ± 1.21 | 42.04 ± 1.32 | 40.69 ± 0.29 | OOM | 43.14 ± 1.22 | 43.23 ± 0.05 | 2.37 ± 0.02 | 3.39 ± 0.09 | OOM | 3.42 ± 4.74 | 5.03 ± 2.95 | 37.99 ± 2.34 | 38.61 ± 1.78 | 24.92 ± 13.49 | 27.48 ± 12.81 |
| | PA | 69.85 ± 3.79 | 67.57 ± 4.72 | 51.80 ± 7.12 | 69.03 ± 2.92 | 68.28 ± 3.63 | 69.78 ± 2.15 | 65.52 ± 5.20 | 68.88 ± 3.34 | 70.83 ± 0.41 | 65.64 ± 1.27 | 67.65 ± 0.26 | 54.87 ± 5.03 | 66.60 ± 4.16 | 68.12 ± 2.49 | 50.05 ± 4.49 | 52.03 ± 3.74 | 54.05 ± 7.21 | 56.19 ± 4.30 |
| | | 52.39 ± 4.16 | 49.24 ± 6.44 | 40.16 ± 6.56 | 51.05 ± 3.63 | 50.03 ± 3.06 | 50.85 ± 4.56 | 55.28 ± 4.97 | 55.13 ± 5.30 | 56.58 ± 5.22 | 53.44 ± 1.52 | 53.59 ± 0.08 | 68.41 ± 3.57 | 48.37 ± 1.78 | 49.26 ± 1.32 | 38.12 ± 3.98 | 39.50 ± 3.60 | 43.79 ± 4.63 | 45.83 ± 3.49 |
| | | 83.35 ± 0.65 | 83.19 ± 0.50 | 66.00 ± 3.71 | 81.43 ± 0.80 | 75.68 ± 0.81 | 79.33 ± 1.00 | 82.04 ± 0.95 | 82.04 ± 0.95 | 84.09 ± 0.65 | 88.35 ± 0.19 | 88.71 ± 0.11 | OOM | 76.99 ± 0.77 | 79.18 ± 0.50 | 65.00 ± 1.60 | 66.01 ± 0.93 | 70.47 ± 2.12 | 71.02 ± 1.72 |
| | | 61.39 ± 1.19 | 61.69 ± 1.33 | 39.92 ± 5.11 | 61.52 ± 0.92 | 60.27 ± 0.66 | 53.48 ± 0.31 | OOT | 63.83 ± 1.04 | 63.93 ± 1.20 | 65.66 ± 0.50 | 65.94 ± 0.32 | OOM | 4.87 ± 14.70 | 7.93 ± 5.29 | 51.41 ± 1.36 | 52.53 ± 0.49 | 36.72 ± 3.12 | 38.29 ± 2.06 |
| | SP | 42.35 ± 1.52 | 35.53 ± 5.14 | 30.69 ± 2.43 | 44.60 ± 2.57 | 39.18 ± 3.79 | 45.77 ± 2.49 | 44.63 ± 1.89 | 44.60 ± 2.57 | 45.85 ± 0.24 | 52.06 ± 2.99 | 54.21 ± 0.36 | 23.04 ± 3.28 | 59.49 ± 2.40 | 60.55 ± 1.14 | 30.74 ± 4.93 | 33.01 ± 2.53 | 35.89 ± 4.60 | 37.19 ± 3.44 |
| | | 26.26 ± 3.22 | 26.89 ± 3.62 | 19.63 ± 2.65 | 25.91 ± 2.88 | 24.13 ± 4.01 | 33.36 ± 1.95 | 26.65 ± 3.12 | 24.82 ± 3.40 | 29.91 ± 0.19 | 48.31 ± 1.91 | 49.68 ± 0.08 | 42.91 ± 2.09 | 39.56 ± 3.09 | 41.39 ± 1.83 | 22.31 ± 3.55 | 24.04 ± 2.49 | 22.14 ± 4.13 | 24.09 ± 2.47 |
| | | 67.41 ± 2.15 | 68.03 ± 1.03 | 51.49 ± 3.49 | 68.18 ± 1.63 | 64.28 ± 1.98 | 63.53 ± 1.36 | OOM | 68.52 ± 1.29 | 69.24 ± 1.19 | 77.91 ± 0.48 | 79.10 ± 0.02 | OOM | 66.65 ± 0.77 | 66.84 ± 0.52 | 48.56 ± 3.63 | 49.92 ± 1.87 | 60.22 ± 2.12 | 61.03 ± 1.89 |
| | | 40.36 ± 1.86 | 39.07 ± 2.43 | 32.82 ± 2.54 | 40.45 ± 2.35 | 38.63 ± 2.28 | 44.88 ± 0.59 | OOM | 39.13 ± 2.16 | 41.22 ± 3.55 | 8.23 ± 2.60 | 26.83 ± 23.95 | OOM | 3.26 ± 4.84 | 5.09 ± 2.99 | 31.45 ± 10.23 | 36.99 ± 5.17 | 11.69 ± 4.86 | 15.95 ± 2.91 |
| Backward | CN | 13.52 ± 1.01 | 14.31 ± 0.49 | 11.70 ± 0.81 | 13.46 ± 1.17 | 11.34 ± 2.84 | 6.30 ± 0.46 | 14.24 ± 0.73 | 14.19 ± 0.46 | 14.49 ± 0.51 | 1.21 ± 0.53 | 2.62 ± 0.14 | OOM | 0.93 ± 0.68 | 1.62 ± 1.37 | OOM | – | 12.83 ± 1.49 | 13.87 ± 0.92 |
| | | 41.88 ± 4.38 | 42.37 ± 4.62 | 31.78 ± 6.07 | 41.27 ± 6.01 | 41.83 ± 3.25 | 47.42 ± 4.67 | 40.67 ± 1.25 | 41.03 ± 5.68 | 43.96 ± 1.18 | 34.70 ± 4.12 | 38.65 ± 0.18 | 38.39 ± 1.43 | 46.04 ± 6.92 | 48.38 ± 4.03 | 42.37 ± 4.88 | 44.09 ± 2.73 | 33.96 ± 6.68 | 35.47 ± 4.71 |
| | | 43.13 ± 5.13 | 40.72 ± 3.60 | 26.36 ± 3.19 | 40.68 ± 2.76 | 38.60 ± 3.79 | 35.68 ± 3.53 | OOT | 39.92 ± 1.09 | 44.87 ± 0.32 | 45.04 ± 2.57 | 46.32 ± 1.02 | 39.85 ± 5.2 | 42.42 ± 5.33 | 44.19 ± 3.89 | 33.35 ± 2.90 | 35.38 ± 1.39 | 30.13 ± 2.87 | 31.93 ± 1.93 |
| | | 28.96 ± 0.77 | 26.77 ± 0.51 | 15.57 ± 2.02 | 27.91 ± 0.41 | 27.24 ± 0.67 | 18.27 ± 1.27 | OOT | 28.67 ± 0.57 | 29.31 ± 0.12 | 22.16 ± 0.66 | 22.43 ± 0.03 | OOM | 21.18 ± 4.10 | 23.10 ± 3.07 | 27.86 ± 0.83 | 28.12 ± 0.57 | 18.33 ± 4.35 | 20.19 ± 2.46 |
| | | 24.16 ± 0.72 | 25.07 ± 0.67 | 21.03 ± 1.57 | 24.40 ± 0.51 | 21.86 ± 0.64 | 23.78 ± 0.23 | OOM | 24.62 ± 0.73 | 25.24 ± 0.01 | 7.18 ± 0.42 | 11.62 ± 5.27 | OOM | 2.40 ± 2.58 | 3.95 ± 1.21 | 12.13 ± 1.06 | 12.81 ± 0.61 | 16.85 ± 5.96 | 18.39 ± 3.89 |
| | PA | 38.68 ± 3.39 | 38.13 ± 3.52 | 16.16 ± 7.56 | 38.33 ± 2.19 | 31.26 ± 4.09 | 36.36 ± 1.12 | 38.01 ± 1.62 | 37.67 ± 2.87 | 39.70 ± 0.26 | 35.30 ± 2.55 | 39.49 ± 0.22 | 36.72 ± 1.89 | 24.19 ± 5.53 | 25.74 ± 3.96 | 37.27 ± 8.62 | 39.84 ± 5.44 | 2.48 ± 1.57 | 3.78 ± 0.89 |
| | | 38.90 ± 1.79 | 38.45 ± 3.22 | 25.10 ± 2.32 | 37.63 ± 1.87 | 37.88 ± 1.11 | 32.83 ± 2.73 | 39.82 ± 2.09 | 38.00 ± 1.24 | 40.07 ± 0.14 | 24.69 ± 5.02 | 26.63 ± 0.10 | 35.84 ± 2.10 | 22.04 ± 7.32 | 24.79 ± 4.82 | 29.30 ± 5.64 | 30.73 ± 3.82 | 43.79 ± 4.63 | 45.19 ± 1.04 |
| | | 28.86 ± 0.97 | 27.51 ± 0.56 | 19.38 ± 4.85 | 28.40 ± 0.74 | 25.25 ± 2.95 | 29.37 ± 0.59 | OOT | 29.04 ± 1.58 | 35.94 ± 4.60 | 22.10 ± 3.30 | 27.05 ± 0.09 | OOM | 15.85 ± 3.96 | 17.12 ± 1.93 | 29.15 ± 1.86 | 30.63 ± 1.29 | 4.74 ± 7.67 | 6.12 ± 5.93 |
| | | 72.45 ± 0.71 | 72.68 ± 0.82 | 9.77 ± 2.30 | 72.45 ± 0.30 | 54.50 ± 3.72 | 29.31 ± 0.74 | OOM | 73.38 ± 0.94 | 59.58 ± 1.12 | 70.66 ± 5.33 | 72.04 ± 0.02 | OOM | 2.40 ± 2.58 | 3.64 ± 1.89 | 79.63 ± 0.82 | 80.01 ± 0.37 | 44.71 ± 26.17 | 45.89 ± 1.58 |
| | SP | 19.30 ± 4.72 | 16.51 ± 6.82 | 11.51 ± 3.65 | 16.98 ± 5.12 | 15.81 ± 2.58 | 28.49 ± 4.19 | 19.02 ± 1.30 | 16.98 ± 5.12 | 22.09 ± 1.10 | 23.95 ± 4.32 | 41.63 ± 0.49 | 23.94 ± 3.68 | 25.47 ± 5.74 | 27.79 ± 4.02 | 11.28 ± 5.03 | 18.12 ± 0.72 | 13.49 ± 4.85 | 15.73 ± 2.81 |
| | | 24.65 ± 3.66 | 27.02 ± 3.20 | 17.81 ± 3.92 | 26.67 ± 3.49 | 25.96 ± 3.86 | 26.58 ± 4.91 | 27.12 ± 2.48 | 26.67 ± 3.49 | 28.25 ± 1.23 | 22.81 ± 2.77 | 24.30 ± 0.93 | 11.74 ± 8.91 | 28.51 ± 3.10 | 29.63 ± 2.91 | 12.98 ± 4.86 | 15.07 ± 1.31 | 17.28 ± 4.22 | 19.17 ± 2.73 |
| | | 22.39 ± 2.29 | 23.05 ± 1.80 | 10.59 ± 3.42 | 22.61 ± 1.73 | 20.92 ± 2.44 | 15.48 ± 1.38 | OOT | 22.61 ± 1.73 | 24.93 ± 0.23 | 23.82 ± 1.54 | 25.44 ± 0.34 | OOM | 25.51 ± 2.29 | 26.05 ± 1.78 | 12.10 ± 2.10 | 14.01 ± 0.29 | 18.27 ± 2.87 | 20.08 ± 1.99 |
| | | 33.50 ± 0.57 | 33.94 ± 0.40 | 33.40 ± 0.94 | 33.48 ± 0.58 | 32.97 ± 0.57 | 34.55 ± 0.53 | OOM | 33.58 ± 0.47 | 33.62 ± 0.49 | 20.99 ± 1.67 | 20.99 ± 1.67 | OOM | 4.17 ± 5.20 | 6.08 ± 4.39 | 19.18 ± 2.46 | 21.09 ± 1.03 | 18.24 ± 12.63 | 19.83 ± 0.83 |
| Real | Collab | 49.49 ± 0.86 | 48.48 ± 1.78 | 44.30 ± 0.61 | 49.35 ± 0.75 | 46.26 ± 1.09 | 50.71 ± 0.21 | OOM | 50.40 ± 1.01 | 52.42 ± 0.08 | 64.83 ± 0.18 | 64.99 ± 0.32 | OOM | – | – | – | – | – | – |
| X-Transfer | Photo → Computers | 87.85 ± 1.92 | 86.68 ± 4.30 | 81.92 ± 1.84 | 86.73 ± 1.31 | 85.83 ± 3.69 | 88.94 ± 1.06 | OOT | 87.48 ± 2.73 | 91.16 ± 1.24 | – | – | OOM | – | – | – | – | – | – |
| X-Transfer | Computers → Photo | 83.96 ± 4.93 | 82.75 ± 3.98 | 82.62 ± 4.57 | 81.94 ± 5.01 | 81.65 ± 4.12 | 86.44 ± 3.15 | OOT | 83.87 ± 5.08 | 91.36 ± 0.05 | – | – | OOM | – | – | – | – | – | – |
| Avg (Δ%) | | -0.02 | -0.89 | -28.84 | -0.58 | -6.85 | -7.09 | -0.19 | – | +5.31 | – | +28.36 | – | – | +21.61 | – | +8.12 | – | +10.08 |

## D  GRAPH GENERATION STATISTICS

Within this section, we further detail how FLEX can generate samples which are link-counterfactual to their training input. As shown within Table 4, we see that the node-degree of FLEX-generated samples more closely-aligns with the testing distribution. However, the clustering coefficient for FLEX-generated samples differs from training for Cora and PubMed but also from testing against all three datasets. Therefore, indicating that FLEX-generated need not fully-align with testing samples in order to improve the baseline GCN performance.

Table 4: Graph Generation Statistics

| Degree | Cora | Citeseer | Pubmed |
|---|---|---|---|
| Train | 3.34 | 3.91 | 4.12 |
| Flex | 2.38 | 2.62 | 2.92 |
| Test | 2.57 | 2.64 | 2.67 |

| Clustering Coefficient | Cora | Citeseer | Pubmed |
|---|---|---|---|
| Train | 0.60 | 0.48 | 0.36 |
| Flex | 0.49 | 0.48 | 0.31 |
| Test | 0.58 | 0.57 | 0.38 |

## E GRAPH GENERATION VISUALIZATIONS

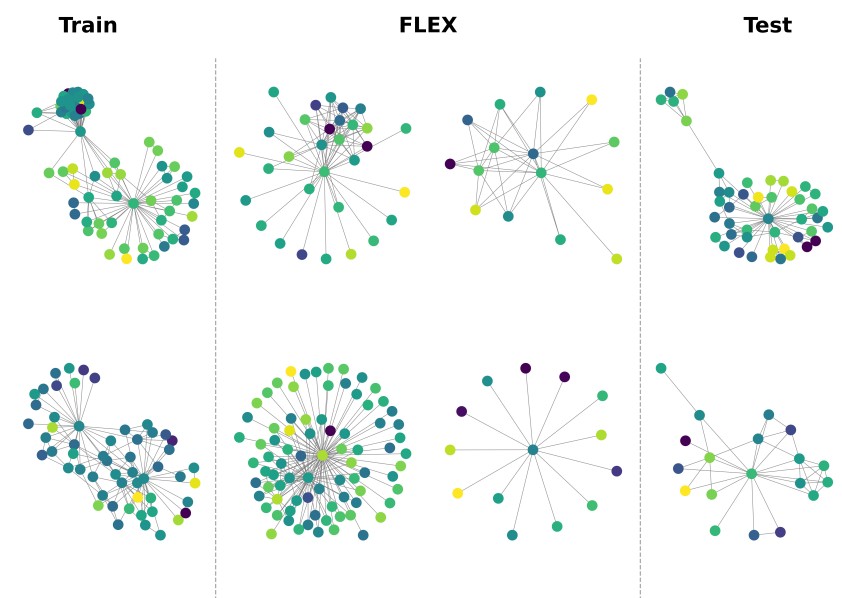

Figure 7: The training, FLEX-generated, and test subgraphs for the ogbl-collab dataset.

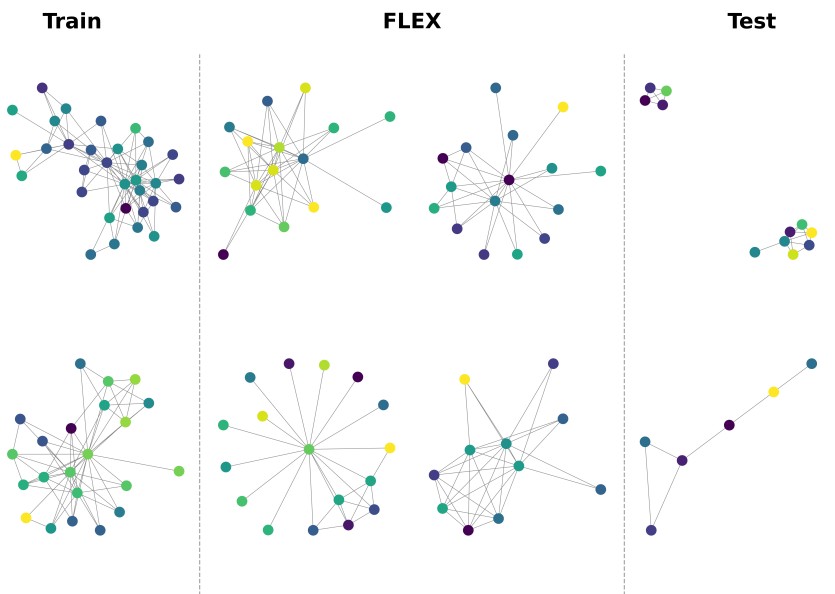

Figure 8: The training, FLEX-generated, and test subgraphs for LPShift's 'Backwards - CN' CiteSeer dataset.

# F    Model Complexity Analysis

Table 5: CFLP edge-calculation pre-processing step with 16 data workers on the "Forward" and "Backward" variants of the LPShift dataset.

| Forward | Cora | CiteSeer | PubMed | ogbl-collab | ogbl-ppa |
|---|---|---|---|---|---|
| CN | 353.88s | 182.33s | 25367.64 s | OOM | OOM |
| PA | 59.7s | 100.7s | 64644.61 s | OOM | – |
| SP | 1282.98s | 634.18s | OOT | OOM | – |
| Backward | Cora | CiteSeer | PubMed | ogbl-collab | ogbl-ppa |
| CN | 2115.47s | 182.33s | 25367.64 s | OOM | OOM |
| PA | 3607.99s | 22932.10s | OOT | OOM | – |
| SP | 625.92s | 36385.35s | OOT | OOM | – |

Table 6: Per-Epoch Training efficiency of FLEX versus CFLP

| Dataset | Models | | | | |
|---|---|---|---|---|---|
| | Cora | CiteSeer | PubMed | ogbl-collab | ogbl-ppa |
| FLEX | 0.366s | 0.450s | 3.19s | 132.7s | 945.2 s |
| CFLP | 0.382s | 0.514s | 56.04s | OOM | OOM |

Table 7: Inference runtime (in seconds) of FLEX versus a baseline GCN across the Common Neighbors Split of the ogbl-collab dataset.

| Dataset | Models | | | | |
|---|---|---|---|---|---|
| | Cora | CiteSeer | PubMed | ogbl-collab | ogbl-ppa |
| GCN | 0.1566s | 0.8839s | 0.4175s | 29.31s | 62.3475s |
| FLEX | 0.1564s | 0.1411s | 0.4207s | 27.4656s | 61.263s |

To verify FLEX's memory and time complexity, we derive the separate components of FLEX's autoencoder and baseline: $L$ = layer, $B$ = batch size, $K$ = noise steps, $e$ = subgraph edge, $n$ = node edge, $d$ = dimension, $d_z$ = sampled dimension, $m$ = candidate samples.

- Encoding works across nodes and edges for: $LB(ed + nd^2)$. Sampling works across the latent dimension for: $KBndd_z$, Decoding works across the final output for: $KBmd_z$. Cumulatively, these three steps work in sequential order for a time-complexity: $T_{batch} = O(LB(ed + nd^2) + KBnd_z + KBmd_z)$.

- FLEX functions with a given GNN backbone ($T_{GNN}$), we abstract this and integrate into the overall framework for a time complexity of: $O(T_{batch} + T_{GNN})$.

- For memory complexity, autoencoders are linear across the sampled latent dimension ($d_z$) on given nodes ($n$) and edges ($m$), where semi-implicit variation aggregates across noise ($K$) to derive: $M_{batch} = O(LB(nd + ed) + KBnd_z)$.

Table 8: The maximum memory (megabytes) utilized when training with of batch size of 32 by a baseline GCN versus GCN integrated within the FLEX framework. Out-of-memory (OOM) occurs on ogbl-ppa due to the severe graph density. In practice, when training on ogbl-ppa we lower the batch size to 4.

| Dataset | Models | | | | |
| --- | --- | --- | --- | --- | --- |
| | Cora | CiteSeer | PubMed | ogbl-collab | ogbl-ppa |
| GCN | 3171.88MB | 3129.93MB | 3653.11MB | 4387.33MB | 5373.76MB |
| +FLEX | 3932.12MB | 3171.88 MB | 3904.29MB | 6387.33MB | OOM |

## G  HYPERPARAMETER SETTINGS

Initial tuning of GCN on all tested datasets and NCN on the LPShift datasets followed a hierarchical approach. Initially, GCN was tuned for 1000 epochs in single runs with early-stopping when validation performance did not improve after 20 steps, a learning rate of $1e-3$ and dropout of 0 across a number of layers $= \{2, 3\}$ and number of hidden channels $= \{128, 256\}$ and batch sizes $= \{32, 64, 128, 256, 512, 1024, 2048, 4096, 8192, 16384, 32768, 65536\}$. Initial NCN tuning followed the same approach, except for being limited to 100 epochs. Dropout and Learning Rate were fixed across the backbone GCN and link predictor.

The second phase of GCN and NCN tuning fixed hidden channels, number of layers, and batch size and then search across a space of learning rate {1e-5, 1e-6, 1e-7} and dropout $= \{0.1, 0.3\}$. NCN was tuned on the ogbl-collab dataset following the author's provided hyperparameters (Wang et al., 2023), as indicated in Table 10. Tuning of the OOD baselines follows the methodology set in (Gui et al., 2022). To do this, we integrate the open-source GOOD (Gui et al., 2022) algorithms within the backbone GCN before feeding the learned GCN embedding to an MLP link-predictor.

To determine the best OOD method hyperparameter settings, we apply the tuned baseline GCN parameters and further tune across OOD loss coefficients as follows: CORAL $= \{0.01, 1.0, 0.1\}$, VREx $= \{10.0, 1000.0, 100.0\}$, IRM $= \{10.0, 0.1, 1.0\}$, DANN $= \{0.1, 1.0, 0.01\}$, GroupDRO $= \{0.01, 1.0, 0.1\}$. Final loss coefficients are shown in Table 9. The number of equal-sized, randomly-sampled environmental subsets were determined in a grid-search across, $e = 3, 4, 5$. The final, $e = 3$ was determined by training loss The number of sampled environmental subsets was fixed at 3 and sampled randomly at program start.

All models, irrespective of FLEX, were evaluated on the full adjacency matrix to ensure consistency with original results.

SIG-VAE, VGAE, and GAE were tuned for 2000 epochs with early stopping set to 100 epochs across learning rates {1e-3, 1e-4}. Models were chosen based on their loss values. All generative auto-encoders were fixed to 32 hidden dimensions and 16 output dimensions to model $\mu$, with variation encoders also modeling $\sigma$. The zero-one labeling trick was applied solely to the generative auto-encoder, with a latent embedding size of $(1000, \text{Num. Hidden})$. Given significant time complexity of pre-training SIG-VAE, a random seed was chosen for SIG-VAE and it's respective GNN and then tested across ten unique seeded runs to obtain final performance.

FLEX was tuned for single seeded runs across learning rates $= \{1e-5, 1e-6\}$ and alpha $= \{0.95, 1.05\}$. Initial sampling runs were tested with threshold values of $= \{0.0, 0.25, 0.5, 0.75, 0.9, 0.99, 0.999, 0.9999\}$,

## H  SYNTHETIC DATASET SPLIT SETTINGS

LPShift datasets were generated following the process described by the authors in (Revolinsky et al., 2024). They consider three types of datasets splits that divide the links based on common heuristics. This includes: *CN* = Common Neighbors (Adamic & Adar, 2003), *SP* = Shortest-Path, *PA* = Preferential-Attachment (Liben-Nowell & Kleinberg, 2003). They further include two "directions" for how the links are split. A 'Forward' splits indicates that the value of the heuristics increase from train to valid and then test. The 'Backwards' split indicates that they decrease. The splits are defined based on two threshold parameters. For the 'Forward' splits the first parameter defines the upper-bound on training data and the second the lower-bound on testing data. The opposite is true for

Table 9: Loss Coefficients for each tested OOD method. $\alpha/\gamma$-threshold for each FLEX-tuned GNN backbone. Ordered from bottom up: Collab, PubMed, Cora, CiteSeer, PPA.

| Dataset | | | Models | | | | | | |
| --- | --- | --- | CORAL | DANN | GroupDRO | VREx | IRM | GCN+FLEX | NCN+FLEX |
| Forward | CN | | 0.01 | 0.01 | 0.1 | 1000.0 | 10.0 | 0.95/0.0 | 1.05/0.0 |
| | | | 0.1 | 0.1 | 0.1 | 100.0 | 0.1 | 0.95/0.0 | 0.95/0.0 |
| | | | 1.0 | 0.1 | 0.01 | 1000.0 | 0.1 | 1.05/0.0 | 0.95/0.0 |
| | | | 0.1 | 0.01 | 0.1 | 10.0 | 0.1 | 0.95/0.0 | 0.95/0.5 |
| | | | 0.1 | 0.01 | 0.1 | 100.0 | 0.1 | 0.95/0.5 | 0.95/0.0 |
| | PA | | 0.01 | 1.0 | 0.1 | 100.0 | 0.1 | 0.95/0.5 | 1.05/0.5 |
| | | | 1.0 | 0.1 | 0.1 | 10.0 | 0.1 | 1.05/0.9 | 0.95/0.0 |
| | | | 1.0 | 0.01 | 0.01 | 10.0 | 0.1 | 0.95/0.5 | 0.95/0.5 |
| | | | 0.1 | 0.01 | 0.1 | 100.0 | 0.1 | 0.95/0.9 | 0.95/0.0 |
| | SP | | 0.1 | 0.01 | 0.01 | 1000.0 | 0.1 | 0.95/0.25 | 0.95/0.25 |
| | | | 1.0 | 1.0 | 0.1 | 10.0 | 0.1 | 0.95/0.0 | 0.95/0.0 |
| | | | 1.0 | 0.01 | 0.01 | 1000.0 | 0.1 | 0.95/0.5 | 0.95/0.0 |
| | | | 0.1 | 0.01 | 0.01 | 100.0 | 1.0 | 0.95/0.5 | 1.05/0.5 |
| Backward | CN | | 0.1 | 0.1 | 0.1 | 1000.0 | 1.0 | 0.95/0.999 | 0.95/0.5 |
| | | | 1.0 | 1.0 | 0.01 | 10.0 | 0.1 | 0.95/0.9 | 0.95/0.99 |
| | | | 0.01 | 0.1 | 0.01 | 100.0 | 0.1 | 0.95/0.9999 | 1.05/0.5 |
| | | | 0.1 | 0.1 | 0.01 | 1000.0 | 0.1 | 0.95/0.99 | 0.95/0.9 |
| | | | 0.1 | 1.0 | 0.01 | 10.0 | 0.1 | 0.95/0.5 | 0.95/0.5 |
| | PA | | 0.01 | 0.01 | 0.1 | 100.0 | 1.0 | 0.95/0.5 | 0.95/0.5 |
| | | | 1.0 | 1.0 | 0.1 | 10.0 | 0.1 | 1.05/0.9 | 0.95/0.5 |
| | | | 1.0 | 0.1 | 0.1 | 10.0 | 0.1 | 0.95/0.9999 | 0.95/0.9 |
| | | | 1.0 | 0.01 | 0.1 | 100.0 | 0.1 | 0.95/0.9 | 0.95/0.9 |
| | SP | | 0.01 | 0.01 | 0.01 | 100.0 | 10.0 | 0.95/0.9999 | 0.95/0.9 |
| | | | 0.1 | 0.01 | 0.1 | 10.0 | 0.1 | 0.95/0.5 | 0.95/0.5 |
| | | | 0.01 | 1.0 | 0.1 | 100.0 | 0.1 | 1.05/0.9999 | 0.95/0.5 |
| | | | 0.1 | 1.0 | 0.01 | 100.0 | 0.1 | 1.05/0.999 | 0.95/0.9 |
| Real | Collab | | 0.1 | 0.01 | 0.1 | 10.0 | 1.0 | 0.95/0.99 | 0.95/0.9 |
| X-Transfer | Photo → Computers | | 1.0 | 0.1 | 0.1 | 100.0 | 10.0 | 0.95/0.99 | 0.95/0.5 |
| X-Transfer | Computers → Photo | | 0.1 | 0.1 | 0.01 | 1000.0 | 10.0 | 0.95/0.9 | 1.05/0.5 |

Table 10: NCN Hyperparameters for the ogbl-collab dataset.

| Parameter | Value | Parameter | Value |
| --- | --- | --- | --- |
| GNN Learning Rate | 0.0082 | Predictor | 0.0037 |
| X Dropout | 0.25 | T Dropout | 0.05 |
| PT | 0.1 | GNN EdgeDropout | 0.25 |
| Predictor Edge Dropout | 0.0 | Predictor Dropout | 0.3 |
| GNN Dropout | 0.1 | Probability Scaling | 2.5 |
| Probability Offset | 6.0 | Alpha | 1.05 |
| Batch Size | 65536 | Layer Norm | True |
| Layer Norm N | True | Predictor | GCN |
| Epochs | 100 | Model | GCN |
| Hidden Dimension | 64 | MP Layers | 1 |
| Test Batch Size | 131072 | Mask Input | True |
| Validation Edges As Input | True | Res. | True |
| Use X. Linear | True | Tail Acting | True |

the 'Backwards' split. For example, the CN split of '1, 2' indicates that training links contain CNs in the range $[0, 1)$, valid in $[1, 2)$, and test $[2, \infty)$. For a CN split of '2, 1', the training and testing links would be flipped. The parameters used across all tested LPShift datasets are detailed below in Table 11 and follow those used by the original authors (Revolinsky et al., 2024). Note that these are the same across all datasets used.

Table 11: LPShift Dataset Parameters.

| 'Backward' Split | Parameters | 'Forward' Split | Parameters |
|---|---|---|---|
| SP | 26, 17 | SP | 17, 26 |
| CN | 2,1 | CN | 1,2 |
| PA | 50, 100 | PA | 100, 50 |

## I  RESOURCES

All models and datasets were tuned and tested on single Nvidia A5000 GPUs with 24 GB available RAM and a server with 128 cores and 1TB available RAM.

## J  HYPERPARAMETER SENSITIVITY

Within this section, we provide further details on the sensitivity analysis conducted on the FLEX framework. As shown in both the 'Backward' and 'Forward' subplots in Figure 9 a higher learning rate contributes to monotonically decreasing performance. This represents a potential pitfall when FLEX-tuning any pre-trained GNNs. Especially since FLEX relies on subgraph samples, whereas GNNs often train on a full adjacency matrix. Within, Figure 10 demonstrates an ablation conducted on the (top) ratio of FLEX-generated subgraphs used for fine-tuning. where a higher-ratio of FLEX-generated subgraphs used within fine-tuning boosts the performance of the GNN backbone by 2%. The (middle) indicates how the number of samples drawn to derive log-likelihood affect performance, with more impact occuring at smaller $J$-values ($< 50$). The (bottom) indicates how the number of $K$ samples for estimating the $\psi$ sampling parameter affects performance, where a pronounced increase occurs where $K < 10$. Figure 10 indicates how the top) ratio of FLEX-generated subgraphs used for fine-tuning the GNN backbone without co-trained parameter-sharing to the generative autoencoder affect performance, where the pre-trained GNN receives an roughly $1\%$ increase from $10\%$ of FLEX-generated subgraphs but limited returns on higher-ratios. The (middle) effect on performance when the ratio of FLEX-generated subgraphs after maximizing KL-divergence is not penalized by a threshold ($\tau$), the limited change indicates how noisy subgraphs obtained from unbounded KL maximization have no capability to boost pre-trained performance. We attribute the significant reduction in performance shown in the final (bottom) image, since the 'from-scratch' trained GNN backbone is unable to distinguish counterfactual links from the original training links.

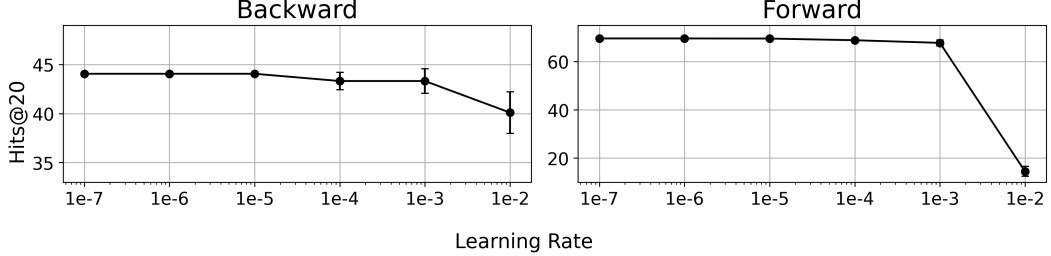

Figure 9: The Hits@20 Scores for FLEX on the "Backwards" - CN CiteSeer Dataset across different learning rates.

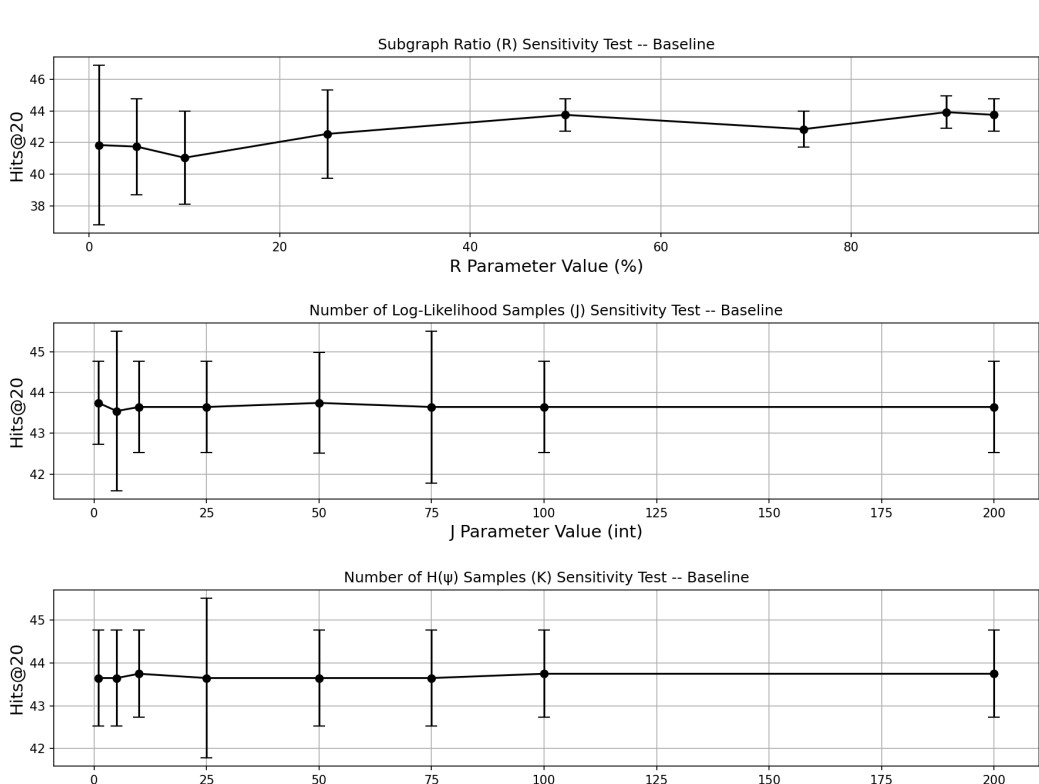

Figure 10: An ablation on the Hits@20 Scores for FLEX on the "Backwards" - CN CiteSeer Dataset, conducted in order: the (top) ratio of FLEX-generated subgraphs used for fine-tuning. The (middle) samples drawn to derive log-likelihood. The (bottom) number of $K$ samples for estimating the $\psi$ sampling parameter. The (third from bottom) ratio of FLEX-generated subgraphs used for fine-tuning the GNN backbone without co-trained parameter-sharing to the generative autoencoder.

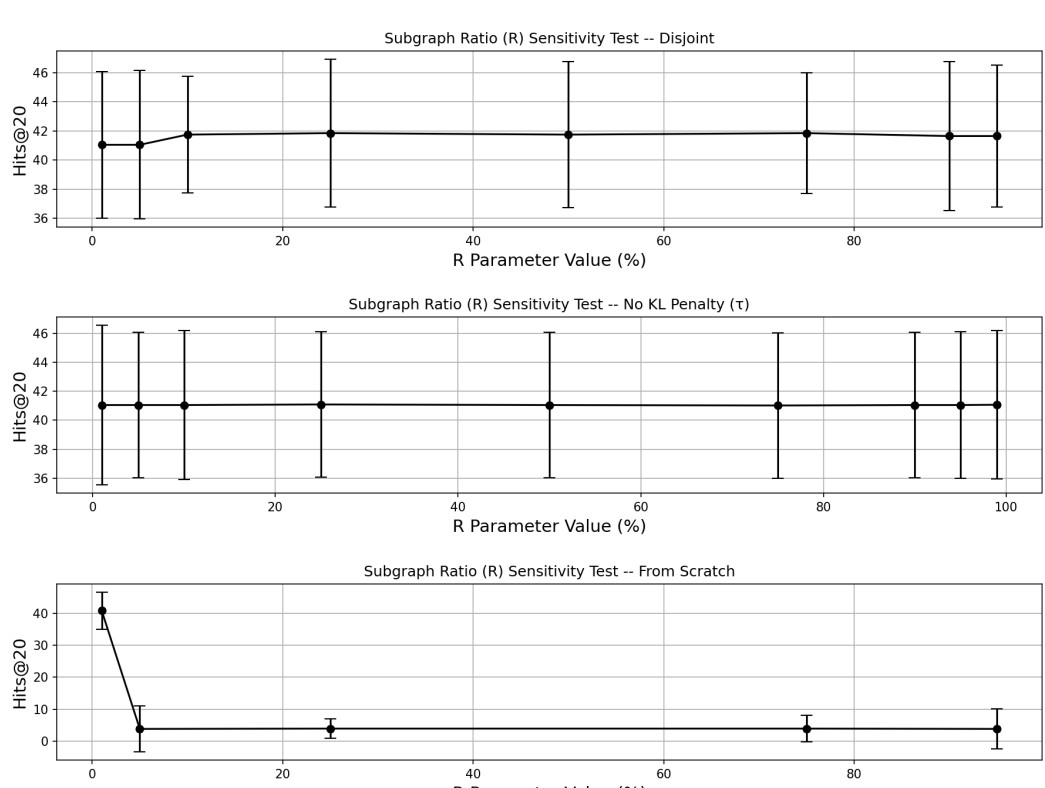

Figure 11: A ablation on the Hits@20 Scores for FLEX on the "Backwards" - CN CiteSeer Dataset, which disconnects the parameters from the generative autoencoder and removes the quadratic penalty, conducted in order: the (top) ratio of FLEX-generated subgraphs used for fine-tuning the GNN backbone without co-trained parameter-sharing to the generative autoencoder. The (middle) ratio of FLEX-generated subgraphs when maximizing KL-divergence is not penalized by a threshold ($\tau$). The (bottom) ratio of FLEX-generated subgraphs when training a GNN 'from-scratch' on FLEX-generated subgraphs with co-trained parameter sharing to the generative autoencoder.

# K GENERATOR ARCHITECTURE

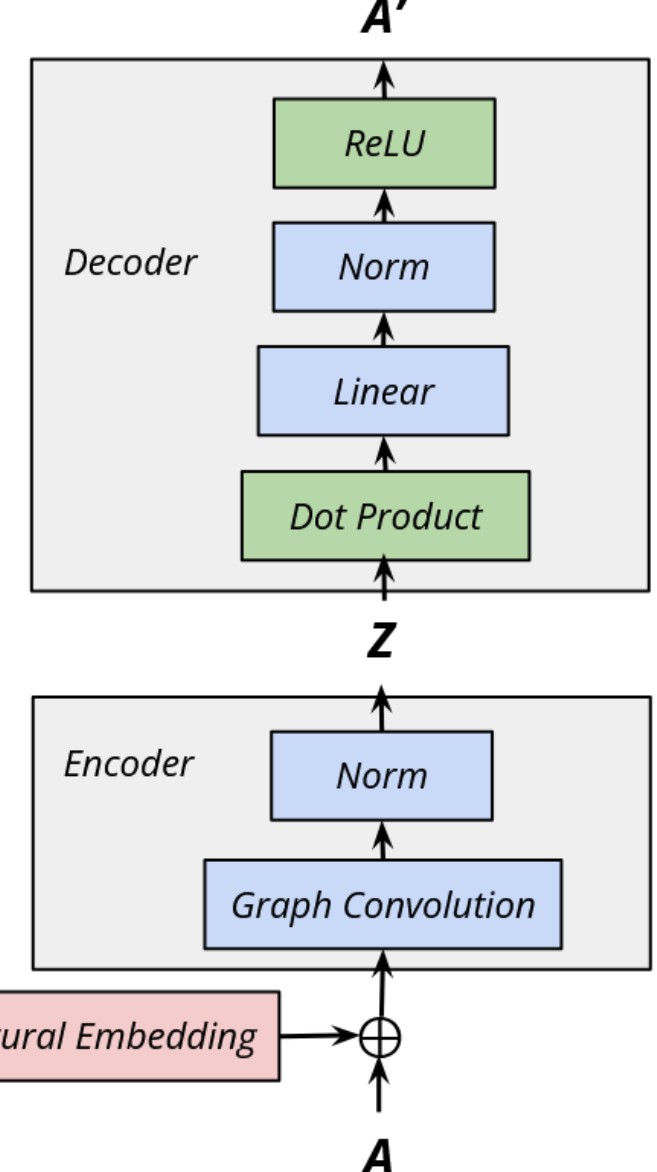

Figure 12: The encoder-decoder module of our proposed framework. Given that training samples are used a direct input, the architecture focuses solely on the block-diagonal adjacency matrix input, $\mathbf{A}$, which is encoded into a learnable latent dimension, $\mathbf{Z}$. An MLP-decoder reads in $\mathbf{Z}$ across target features to output the augmented subgraph, $\mathbf{A}'$. The MLP-decoder can be swapped for the original Bernoulli-Poisson decoder, proposed in Hasanzadeh et al. (2019).

## L  FLEX ALGORITHMS

As defined earlier in Section 3, FLEX operates in two critical stages, (1): The generative graph model (GGM) is pretrained on labeled subgraphs extracted from the target dataset following Eq. equation 2. While the GNN is pre-training separately on the full adjacency matrix. This is defined on lines 3-5 in Algorithm 1. (2): After pre-training, the generative GGM is then placed within the FLEX framework and co-trained with the GNN following Eq. equation 8. At each subsequent mini-batch, the GGM produces new synthetic graphs and therefore new structural views of the original dataset which are subsequently passed into the GNN to gauge sample validity. This is defined on lines 6-10 in Algorithm 1. Given that the divergence between the posterior and prior distributions is maximized, this means that subsequent epochs should converge to generate a final distribution that is structurally different from the training samples. As mentioned in Section 3.3.1, Algorithm 2 takes in feature input and a representative block-diagonal matrix to ensure that SIG-VAE is expressive to mini-batch samples of varying node numbers (Hasanzadeh et al., 2019).

---

**Algorithm 1** FLEX - Pre-training and Tuning

---

**Require:** $\mathbf{G}(\mathbf{X}, \mathbf{A})$, $\mathbf{X} \in \mathbb{R}^{N \times d}$
1: Extract $\mathbf{G_s}$ from 1-hop enclosed subgraphs of $A$
2: Retrieve $\mathbf{Z}$ using the zero-one labeling trick, Eq. equation 6
3: **for** epoch = 1 to pretrain **do**
4:     Train SIG-VAE on $\mathbf{G_s}$ using Eq. equation 2 and labels $\mathbf{Z}$
5: **end for**
6: **for** epoch = 1 to flex-tune **do**
7:     Sample $\mathbf{G'_s}$ from SIG-VAE
8:     Apply Eq. equation 6 on $\mathbf{G'_s}$
9:     Train GNN + SIG-VAE on $\mathbf{G'_s}$ using Eq. equation 8
10: **end for**

---

**Algorithm 2** Node-Aware Decoder Algorithm

---

**Require:**
    $x \in \mathbb{R}^{N \times F}$: Node features, $A = \text{diag}(A_1, \ldots, A_K)$: Block-diagonal adjacency
    $\mathbf{Z} \in \mathbb{R}^{N \times d}$: Structural features, $J$: Truncation index, $n_{train} = [\mathcal{N}_1, \ldots, \mathcal{N}_N]$: Training Nodes

1: $(\mu, \log \sigma^2, \text{SNR}) \leftarrow \text{Encoder}(x, A, Z)$
2: $\mu' \leftarrow \mu_{J:N}, \quad \log \sigma'^2 \leftarrow \log \sigma^2_{J:N}$
3: Split $\mu', \log \sigma'^2$ into subgraphs $\mu_i, \log \sigma_i^2$ using $n_{train}$
4: **for** $i = 1$ to $N$ **do**
5:     Sample $\epsilon_i \sim \mathcal{N}(0, I)$
6:     $z_i \leftarrow \mu_i + \epsilon_i \odot \exp(0.5 \cdot \log \sigma_i^2)$                    ▷ Reparametrization Trick
7:     $(\hat{A}_i, z_i^{\text{scaled}}, r_k) \leftarrow \text{Decoder}(z_i)$
8:     Insert $\hat{A}_i$ into $\hat{A}_{\text{global}}$ at block $(i, i)$
9:     Insert $z_i, z_i^{\text{scaled}}, \epsilon_i$ into global tensors
10: **end for**
11: **return** $\hat{A}_{\text{global}}, \mu, \log \sigma^2, \mathbf{Z}_{\text{global}}, \mathbf{Z}_{\text{global}}^{\text{scaled}}, \epsilon_{\text{global}}, r_k, \text{SNR}$

---

## M  DEGREE BIAS INVESTIGATION

As previously-mentioned in Section 3.3.1, the generated subgraph samples without an indicated threshold suffer from degree-bias (Tang et al., 2020), thereby resulting in densely-generated outputs, even on sparse inputs. This effect is demonstrated in Figure 13, as shown with the perfect linear relationship between the mean number of common neighbors in the output sample respective to the number of nodes within input samples. To combat this, the indicator threshold is tuned to eliminate edge-probabilities with a lower threshold than indicated. The effect of this threshold can be seen in Figure 14, where a threshold of 0.9999 reduces the maximum mean number of Common Neighbors

by a factor of 40, as respective to Figure 13. This then shows a more meaningful correlation between output CNs and input nodes, meaning that output graphs are no longer densely-connected which serves as a desirable property when attempt to generalize on much sparser graphs; like those contained within the 'Backward' CN Cora dataset.

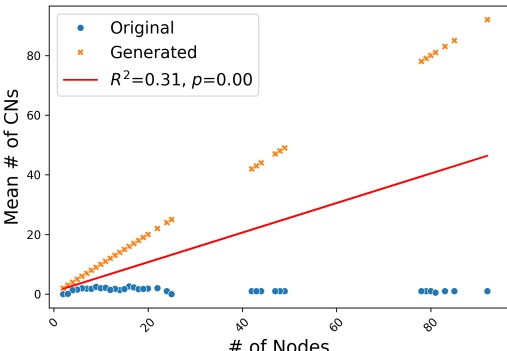

Figure 13: The distribution of Mean Common Neighbors and Mean Number of Nodes for subgraph samples generated by FLEX on the 'Backward' LPShift CN - Cora dataset without the threshold function. Note the near-perfect linear growth of Common Neighbors with respect to the number of nodes within a given input subgraph.

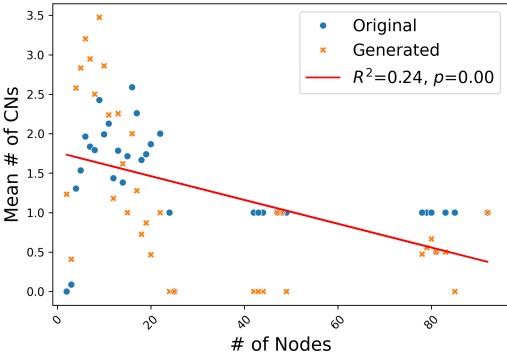

Figure 14: The distribution of Mean Common Neighbors and Mean Number of Nodes for subgraph samples generated by FLEX on the 'Backward' LPShift CN - Cora dataset after applying the threshold function. The threshold function ensures that low-probabilities edges are not formed, resulting in generated samples with a common neighbors that are morely closely-correlated to the input samples.

## N  DATASET LICENSES

Both OGB (Hu et al., 2020) and LPShift (Revolinsky et al., 2024), the datasets considered in our study, are licensed under the MIT license.

## O  LIMITATIONS

From a theoretical perspective, FLEX operates under the critical assumption that there are counter-factual substructures which exist under the causal model that constructed the original dataset. If no such substructures are present, (i.e. the dataset samples are not OOD), then FLEX is also likely to decrease model performance.

For practical implementation, FLEX requires sampling $k$-hop enclosed subgraphs, which can be computationally-restrictive if applied with the same settings as training on full adjacency matrices. Additionally, if poorly-tuned, then SIG-VAE will produce meaningless outputs and decrease down-stream performance regardless of how well pre-trained the GNN is. FLEX has a high-likelihood of

inducing dataset drift, where a single epoch can increase performance but subsequent epochs will likely lead to a monotonic decrease in performance.

This work introduces, formalizes, and demonstrates the notion that **it is possible** to generate counterfactual link-structure and then apply those same structures to improve OOD performance. It does not claim to fully-understand this mechanism but instead bring awareness to a phenomena that can elevate the performance of current link-prediction models and their robustness to OOD data.

## P SOCIETAL IMPACT

Our proposed method, FLEX, aims to improve the generalization capabilities of link prediction methods. Since generalization is a key real-world concerns for many ML models, we argue that FLEX has a potential to have a positive impact. Furthermore, link prediction is a common task used in many fields such as recommender systems, drug-drug interactions, and knowledge graph reasoning. Thus, improving the generalization of link prediction in those fields can be helpful for future research. Therefore, no apparent risk is related to the contribution of this work.

