# OpenReview forum: "Subgraph Generation for Generalizing on Out-of-Distribution Links"
_ICLR.cc/2026/Conference — Submitted to ICLR 2026_

### Official Review · Reviewer_Xvsx · 2025-10-29

**Soundness:** 3
**Presentation:** 3
**Contribution:** 2
**Rating:** 4
**Confidence:** 4

**Summary:**

The paper tackles OOD generalization for link prediction. The core idea is to augment training with counterfactual subgraphs synthesized by a generative model so that a pre‑trained link predictor becomes robust to structural shifts. The proposed framework, FLEX, first pretrains a GNN and a semi‑implicit variational autoencoder on k‑hop subgraphs extracted with the labeling trick, then co‑trains them in an adversarial way (much like GAN). Counterfactuality is encouraged by maximizing KL divergence between the posterior and prior. On the LPShift benchmark, the method improves over baselines in most cases. Efficiency analyses show FLEX training/inference overheads are modest compared to a non‑parametric counterfactual baseline like CFLP.

**Strengths:**

1. It has a well‑motivated problem and clear intuition.

2. FLEX has simple, modular setup. It can be used with any pre‑trained GNN. and the $$/gamma$$ threshold tackle the tendency of generators to over‑densify.

3. The empirical results over LPShift show the effectiveness of the FLEX.

**Weaknesses:**

1. Limited backbone and generator diversity. Results are only on GCN and NCN. Modern link‑predictors (e.g., NBFNet, Neo‑GNN, LPFormer) are referenced but not evaluated; likewise, the framework is described as agnostic to the generator, but only SIG‑VAE is used. A “FLEX‑with‑VGAE/diffusion” variant or at least a plug‑in ablation would isolate how much the semi‑implicit choice matters.

2. I am concerned about the efficiency. The recent trend of modern LP models has been shifted from subgraph-level prediction (SEAL) to more efficient node-level encoding (BUDDY, NCN). This shift makes the method efficient and applicable to the real-world use case with large-scale graphs. However, FLEX still operates at subgraph-level, meaning that it will struggle to scale to large graphs. For example, OGBL-Citation2 can be a good test bed to evaluate FLEX on large-scale dataset.

**Questions:**

1. During inference time, is the cotrained GNN predictor being used for prediction or a new GNN being trained from scratch on the original graph+FLEX-generated graphs? If the former, will FLEX, as a data augmentation method, generalize across different GNN backbones? In other words, if the FLEX-generated graphs can improve performance of any LP methods (including both train-from-scratch GNN or even just heuristics like Common Neighbor), it will have much broader use case.


2. Can FLEX improve the model performance not only on the distribution shift dataset but also general ones (which already has some degree of distribution shift)?

---

> ### Author Response · Authors · 2025-11-24
> **Response to Reviewer Xvsx (1/2)**
>
> 1. "Limited backbone and generator diversity. Results are only on GCN and NCN. Modern link‑predictors (e.g., NBFNet, Neo‑GNN, LPFormer) are referenced but not evaluated".
> * We apply additional tests with GAT, GIN, and HL-GNN to indicate that FLEX functions across additional backbones. We select HL-GNN given it's recent addition as a link predictor model [1].
>
> **shown below** (Table 1 -- HIts@20 scores for additional tests using base models, and FLEX-tuned (+FLEX) models.)
> | Direction | Type | HL-GNN | HL-GNN+FLEX | GAT | GAT+FLEX | GIN | GIN+FLEX |
> |-----------|------|--------|-------------|-----|----------|-----|----------|
> | Forward | CN | 36.16 ± 3.30 | 37.81 ± 2.71 | 41.28 ± 2.82 | 42.43 ± 2.34 | 41.43 ± 7.56 | 43.50 ± 5.97 |
> | Forward | PA | 49.21 ± 5.35 | 51.12 ± 2.86 | 51.15 ± 2.86 | 52.52 ± 2.19 | 51.26 ± 4.27 | 52.83 ± 2.89 |
> | Forward | SP | 42.24 ± 2.60 | 43.47 ± 1.62 | 33.27 ± 5.58 | 35.99 ± 3.01 | 32.48 ± 3.93 | 34.56 ± 2.68 |
> | Backward | CN | 22.59 ± 3.92 | 24.24 ± 2.71 | 28.93 ± 2.42 | 30.10 ± 1.33 | 22.42 ± 4.27 | 23.97 ± 2.78 |
> | Backward | PA | 16.12 ± 4.85 | 17.82 ± 3.90 | 43.84 ± 4.23 | 45.30 ± 2.73 | 23.93 ± 10.01 | 25.24 ± 2.36 |
> | Backward | SP | 20.92 ± 4.08 | 22.39 ± 3.68 | 13.89 ± 3.61 | 17.07 ± 0.84 | 16.82 ± 6.14 | 18.70 ± 2.09 |
> | -- | **Δ%** | -- | **+6.04%** | -- | **+7.34%** | -- | **+6.35%** |
>
> [1] Zhang, Juzheng, Lanning Wei, Zhen Xu, and Quanming Yao. "Heuristic learning with graph neural networks: a unified framework for link prediction." In Proceedings of the 30th ACM SIGKDD conference on knowledge discovery and data mining, pp. 4223-4231. 2024.
>
> 2. "likewise, the framework is described as agnostic to the generator, but only SIG‑VAE is used. A “FLEX‑with‑VGAE/diffusion” variant or at least a plug‑in ablation would isolate how much the semi‑implicit choice matters."
> * We clarify that our original submission's Table 2 does include a plug-in ablation which compares FLEX to a 'FLEX-with-VGAE' architecture. We note Table 2 indicates a 2-10% drop when applying VGAE as generator versus SIGVAE.
>
> 3. "I am concerned about the efficiency. The recent trend of modern LP models has been shifted from subgraph-level prediction (SEAL) to more efficient node-level encoding (BUDDY, NCN). This shift makes the method efficient and applicable to the real-world use case with large-scale graphs. However, FLEX still operates at subgraph-level, meaning that it will struggle to scale to large graphs. For example, OGBL-Citation2 can be a good test bed to evaluate FLEX on large-scale dataset."
> * FLEX uses subgraphs to overcome graph generation scalability bottlenecks, as demonstrated with our tests on OGBL-PPA (6x the edge density of OGBL-CITATION2). SOTA graph generative models rarely scale beyond 5000 nodes, as shown by 'OOM' results in Bergmeister et. al's work [1].
> *  Due to the design of BUDDY, successful integration into FLEX would be prohibitively expensive. BUDDY calculates the subgraph hashes for each link. To avoid efficiency concerns, BUDDY does this as a preprocessing step before training or evaluation. However, for FLEX, we are constantly generating new subgraphs. So, BUDDY would require re-calculating subgraph hashes on new and intermediate graphs before FLEX even tunes a backbone GNN; which is prohibitive for sampling across multiple noisy steps in graph generation.
>
> [1] Bergmeister, Andreas, Karolis Martinkus, Nathanaël Perraudin, and Roger Wattenhofer. "Efficient and Scalable Graph Generation through Iterative Local Expansion." In The Twelfth International Conference on Learning Representations.

---

> ### Author Response · Authors · 2025-11-24
> **Response to Reviewer Xvsx (2/2)**
>
> 4. "During inference time, is the cotrained GNN predictor being used for prediction or a new GNN being trained from scratch on the original graph+FLEX-generated graphs? If the former, will FLEX, as a data augmentation method, generalize across different GNN backbones? In other words, if the FLEX-generated graphs can improve performance of any LP methods (including both train-from-scratch GNN or even just heuristics like Common Neighbor), it will have much broader use case."
> * Thank you for the engaging question! Yes, the co-trained GNN is currently used for prediction within "GCN+FLEX" and "NCN+FLEX" tests. We treat this co-training as analogous to the fine-tuned decoding phase of CFLP [1]. In practice, the co-training is critical to ensure the GNN backbone separates incoming counterfactual edges from the original learned distribution. Table 2 demonstrates this requirement, where we train a GCN from scratch with increasing ratios of FLEX-to-original subgraphs. Notably, FLEX-generated subgraphs exceeding a 5% ratio causes GNN performance to plummet from 41% to roughly 4%.
>
> **shown below** (Table 2 -- The Hits@20 impact of raising the ratio of FLEX subgraphs used for training a GNN from scratch. A sharp decrease in performance occurs once FLEX subgraphs compose 5% of total samples.)
> | Percentage of FLEX subgraphs | 0% (Original) | 1% | 5% | 10% | 25% | 50% | 75% | 90% | 95% | 100% |
> |------|------|------|------|------|------|------|------|------|------|------|
> | Final Test | 41.03 ± 5.68 | 33.13 ± 5.94 | 3.82 ± 7.80 | 3.96 ± 5.17 | 3.96 ± 7.24 | 3.96 ± 4.97 | 3.96 ± 4.09 | 3.96 ± 3.02 | 3.96 ± 6.24 | 3.96 ± 6.17 |
>
> 5. "Can FLEX improve the model performance not only on the distribution shift dataset but also general ones (which already has some degree of distribution shift)?"
> * Yes, our original submission results within Table 1 and 3 includes a 2% increase in Hits@50 for the real-world ogbl-collab, and 3-7% increase in AUC for domain-transfer between Amazon-Photos and Amazon-Computers, indicating that FLEX functions on real-world dataset shift.

---

### Official Review · Reviewer_NgWx · 2025-10-30

**Soundness:** 2
**Presentation:** 2
**Contribution:** 2
**Rating:** 4
**Confidence:** 5

**Summary:**

This paper introduces FLEX, a framework designed to improve OOD generalization in GNNs for the link prediction task. The core idea is to jointly train a GGM and a GNN to generate counterfactual subgraphs that expand the structural support of the training distribution. The framework demonstrates considerable innovation and empirical effectiveness, achieving efficient subgraph-level generation through a well-designed training mechanism. By reformulating link prediction in terms of structural feature distributions (e.g., Common Neighbors), the paper provides a principled theoretical foundation for understanding the limitations of traditional link predictors under distribution shift. Extensive experiments on multiple benchmark datasets demonstrate that FLEX outperforms both traditional OOD baselines and graph-specific methods in robustness and generalization performance.

**Strengths:**

By reformulating the task in terms of structural feature distributions (e.g., Common Neighbors), the paper provides a principled explanation for why traditional link predictors underperform under distribution shifts. The proposed set-theoretic and ELBO-based analysis forms the unified theoretical perspectives on OOD generalization for link prediction.

The empirical evaluation spans multiple benchmark datasets and diverse graph structures, consistently demonstrating the robustness and OOD generalization ability of FLEX on link prediction tasks.

FLEX performs counterfactual generation at the subgraph level, which is an efficient and scalable design choice. This approach reduces unnecessary graph-wide computation and provides a practical path toward efficient OOD generalization.

**Weaknesses:**

Although the appendix provides a set-theoretic argument that counterfactual subgraph generation can enlarge the overlap between training and testing distributions, the analysis remains qualitative. It lacks a quantitative derivation of generalization bounds, risk functions, or error guarantees. The theoretical foundation that KL divergence regularization and structural diversity objectives necessarily improve OOD generalization remains insufficient.

The paper emphasizes generating “structurally different” counterfactual subgraphs, yet it does not explain how these generated subgraphs maintain semantic or structural validity. Furthermore, no visualization or statistical characterization of the generated samples is provided.

All experiments are conducted on homogeneous graphs for the link prediction task, its applicability to heterogeneous or more complex graph structures remains unverified.

The paper does not sufficiently isolate the contribution of each component within FLEX, as it lacks sensitivity analysis on the number and perturbation strength of generated subgraphs, ablation results for removing the KL constraint, and comparison between joint training with GNNs and independent optimization, making it difficult to determine whether the observed improvements truly originate from the proposed mechanism rather than from generic data augmentation or increased model capacity.

The paper claims efficiency through subgraph-level generation, yet several experiments report OOM or training exceeding 24 hours. There is no analysis of computational complexity, runtime, or hardware requirements.

Key implementation details such as generator architecture, sampling strategy, and loss coefficients are missing.

The presentation is weak, with numerous grammatical, stylistic, and typographical errors throughout the manuscript. For example, line 178 “it’s”; line 240 “said features”; and line 340 “we an input links…”.

**Questions:**

This paper provides a set-theoretic argument that counterfactual subgraph generation can enlarge the overlap between training and testing distributions. However, this analysis is mostly qualitative and lacks quantitative characterization of the generalization error or risk bounds. Could the authors provide a more rigorous theoretical or empirical analysis to substantiate the claimed generalization improvement?

Could the authors provide visualization or statistical characterization of the generated subgraphs to demonstrate their structural diversity and semantic consistency?

All experiments are conducted on homogeneous, static graphs for the link prediction task. Have the authors evaluated FLEX on more complex graph settings such as heterogeneous graphs (e.g., MAG [1], DBLP [2])? If not, could they discuss the applicability or potential limitations of the proposed framework under such conditions?

The paper lacks a thorough ablation study to isolate the contribution of each FLEX component. Could the authors supplement experiments that (1) analyze the sensitivity to the number and perturbation strength of generated subgraphs, (2) evaluate performance without the KL-divergence constraint, and (3) compare joint versus independent training of FLEX and GNNs?

Several experiments are marked as “OOM” or “>24h,” but no analysis of runtime, complexity, or hardware requirements is provided. Could the authors include a detailed analysis of computational complexity, training time, memory consumption, and hardware setup to clarify FLEX’s scalability and practical feasibility?

The paper omits important implementation details such as the generator architecture, sampling strategy, and loss coefficients. To enhance reproducibility, could the authors release the implementation and provide detailed hyperparameter settings and training configurations?

[1] Hu, et al. OGB-LSC: A Large-Scale Challenge for Machine Learning on Graphs. NeurIPS 2021.
[2] Zhang, et al. Oag-bench: a human-curated benchmark for academic graph mining. KDD 2024.

---

> ### Author Response · Authors · 2025-11-24
> **Response to Reviewer NgWx (1/2)**
>
> 1. "This paper provides a set-theoretic argument that counterfactual subgraph generation can enlarge the overlap between training and testing distributions. However, this analysis is mostly qualitative and lacks quantitative characterization of the generalization error or risk bounds. Could the authors provide a more rigorous theoretical or empirical analysis to substantiate the claimed generalization improvement?"
>
> * Thank you for the suggestions, we update Appendix Section B with Proposition B.1, Corollary B.3, and Remark 4; formalizing the intuiton about how expanded coverage reduces OOD risk.
>     * Specifically, Proposition B.1 establishes coverage-based risk bounding, where $R_{test}(f)≤R_{train}′(f)+δ′$, where $δ′=P_{tes}(U∖T′)$ measures the remaining test mass outside the augmented training support.
>     * Corollary B.3 links Theorem 1 and Lemma 1 to show that successful FLEX-tuning enlarges the training–test overlap in representation space and reduces estimated risk, $δ′<δ$.
>     * Remark 4 further clarifies how the KL-structural diversity objective affects risk reduction by increasing Lebesgue-measure overlap $μ(S∩(U∖T))$.
>
> 2. "Could the authors provide visualization or statistical characterization of the generated subgraphs to demonstrate their structural diversity and semantic consistency?"
>
> * The new Figures 7 and 8 in Appendix E, shows the visible change in graph structure between train, test, and Flex-generated subgraphs. This further aligns with graph generation statistics in Appendix H and the revised theory in Appendix B; indicating rigorous theoretical and empirical proof of how FLEX generates counterfactual subgraphs to improve risk bounds and enhance downstream performance.
>
> 3. "All experiments are conducted on homogeneous, static graphs for the link prediction task. Have the authors evaluated FLEX on more complex graph settings such as heterogeneous graphs (e.g., MAG [1], DBLP [2])? If not, could they discuss the applicability or potential limitations of the proposed framework under such conditions?".
>
> We appreciate the engaging question! We make a few distinctions about FLEX in heterogenous LP and our research objective:
> *  FLEX's core tactic is maximizing structurally-conditioned KL-divergence to generate a non-marginal overlap between OOD train and testing feature distributions. If train and target links are too dissimilar (i.e. a wide margin in non-overlapping feature distributions), generated-subgraphs degenerate into noise; likely reducing downstream performance.
> * Our tests include the original ogbl-collab, which is a subset of the cited MAG dataset; specifically constructed for link-prediction instead of node classification [3].
> *  The DBLP and MAG datasets were originally designed for heterogenous node classification [4]. The proper adaption of MAG and DBLP for large-scale heterogenous link-prediction within FLEX requires extensive research beyond the available time and single-GPU constraints for our current testing system.
> *   We acknowledge that overlap between the heterogenous and OOD problems within link-prediction is an exciting research direction. However, our rebuttal's literature review shows no precedent for the direct use of generative graph autoencoders within heterogenous link-prediction [5, 6]. This lack of precedent means successfuly merging heterogenous LP into our current submission will require foundational experiments. Such a deep level of experimentation would distract from our original argument of parametrizing counterfactual learning to improve OOD performance in link prediction.
>
> [3] https://ogb.stanford.edu/docs/linkprop/#dataset-ogbl-collab-leaderboard
>
> [4] Fu, Xinyu, Jiani Zhang, Ziqiao Meng, and Irwin King. "Magnn: Metapath aggregated graph neural network for heterogeneous graph embedding." In Proceedings of the web conference 2020, pp. 2331-2341. 2020.
> [5] Tian, Yijun, Kaiwen Dong, Chunhui Zhang, Chuxu Zhang, and Nitesh V. Chawla. "Heterogeneous graph masked autoencoders." In Proceedings of the AAAI conference on artificial intelligence, vol. 37, no. 8, pp. 9997-10005. 2023.
> [6] Li, Xiang, Tiandi Ye, Caihua Shan, Dongsheng Li, and Ming Gao. "Seegera: Self-supervised semi-implicit graph variational auto-encoders with masking." In Proceedings of the ACM web conference 2023, pp. 143-153. 2023.

---

> ### Author Response · Authors · 2025-11-24
> **Response to Reviewer NgWx (2/2)**
>
> 4. "The paper lacks a thorough ablation study to isolate the contribution of each FLEX component. Could the authors supplement experiments that (1) analyze the sensitivity to the number and perturbation strength of generated subgraphs, (2) evaluate performance without the KL-divergence constraint, and (3) compare joint versus independent training of FLEX and GNNs?"
>
> * Thank for you for the suggestion, within our revised Appendix Section J, Figures 10 and 11, we have included an additional sensitivity analysis across:
>     *  (Figure 10, top) shows how a higher ratio of FLEX-generated subgraphs improve GNN backbone performance.
>     *  (Figure 10, middle and bottom) shows the relative effectiveness of semi-implicit sampling parameters at lower K and J values.
>     *  (Figure 11, top), shows the necessity of joint-training to enable counterfactual understanding in the fine-tuned GNN.
>     *  (Figure 11, middle), indicates that unbounded KL-maximization leads to noisy subgraphs which do not improve pre-trained performance.
>
> 5. "Several experiments are marked as “OOM” or “>24h,” but no analysis of runtime, complexity, or hardware requirements is provided. Could the authors include a detailed analysis of computational complexity, training time, memory consumption, and hardware setup to clarify FLEX’s scalability and practical feasibility?"
> * We clarify that "OOM" or "OOT/ >24h" values within the original paper's Table 1, 3, 5 and 6 **is for CFLP, not FLEX**.
> * Our original submission evaluates: training time, inference runtime, memory consumption, and hardware setup within Table 5, 6, 7, 8, and Appendix I, respectively.
> * To verify FLEX's memory and time complexity, we derive the separate components of FLEX's autoencoder and baseline: $L$ = layer, $B$ = batch size, $K$ = noise steps, $e$ = subgraph edge, $n$ = node, $d$ = dimension, $d_z$ = sampled dimension, $m$ = candidate samples.
> * Encoding works across nodes and edges for: $L · B · (e·d + n · d^2)$. Sampling works across the latent dimension for: $K · B · n · d · d_z$, Decoding works across the final output for: $K · B · m · d_z$. Cumulatively, these three steps work in sequential order for a time-complexity: $T_{batch} = O(L · B · (e·d + n· d^2) + K · B · n · d_z + K · B · m · d_z )$.
> * FLEX functions with a given GNN backbone ($T_{GNN}$), we abstract this and integrate into the overall framework for a time complexity of: $O(T_{batch} + T_{GNN})$.
> * For memory complexity, autoencoders are linear across the sampled latent dimension ($d_z$) on given nodes ($n$) and edges ($e$), where semi-implicit variation aggregates across noise ($K$) to derive: $M_{batch} = O(L · B · (n·d + e·d) + K · B · n · d_z)$
> * Our theoretical verification of memory and runtime complexity is included in Appendix F of the revised submission.
>
> 6. "The paper omits important implementation details such as the generator architecture, sampling strategy, and loss coefficients. To enhance reproducibility, could the authors release the implementation and provide detailed hyperparameter settings and training configurations?"
> * We clarify there is a reference implementation attached within the supplementary material of this submission as a zip file.
> * We agree that implementation details are important for reproducibility:
>     *  Our new Appendix Section K, Figure 12 includes an architectural diagram with individual implementation components for our SIG-VAE graph generation architecture.
>     *  We add Table 9 to Appendix G, 'Hyperparameter Settings' to show the best loss coefficients across FLEX and other tested OOD architectures.
>     *  For sampling, FLEX selects samples from the reparametrized embedding based on log-likelihood and $\psi$ (learned by respective parameters J and K), which is then directly-decoded into graph samples [1].
>
> [1] Hasanzadeh, Arman, Ehsan Hajiramezanali, Krishna Narayanan, Nick Duffield, Mingyuan Zhou, and Xiaoning Qian. "Semi-implicit graph variational auto-encoders." Advances in neural information processing systems 32 (2019).
>
> 7. "The presentation is weak, with numerous grammatical, stylistic, and typographical errors throughout the manuscript. For example, line 178 “it’s”; line 240 “said features”; and line 340 “we an input links…”."
> * Thank you for bringing this to our attention, we have revised the grammar in our submission to enhance presentation.

---

> > ### Comment · Reviewer_NgWx · 2025-11-27
> >
> > Thank you for the rebuttal. However, many issues—such as the theoretical analysis, ablation studies, and complex-graph evaluations—remain unresolved, and several revisions mentioned in the response (e.g., Proposition B.1, Corollary B.3, Remark 4, Figure 12) do not appear in the updated manuscript. I therefore maintain my original score.

---

> ### Author Response · Authors · 2025-11-27
>
> Our apologies! We forgot to upload the revised manuscript.
>
> The manuscript has now been updated with all the stated revisions. All revisions are highlighted in blue.
>
> Sorry again for the confusion.

---

> > ### Author Response · Authors · 2025-12-03
> >
> > In conclusion, we would like to clarify that the updated manuscript answers the following concerns:
> > * Theoretical Analysis
> > * Ablation Study
> > * Proposition B.1, Corollary B.3, Remark 4, and Figure 12
> >
> > In addition, we have conducted tests which apply FLEX to GCN on the DBLP dataset [1]. We note that DBLP was previously-designed for node classification and node clustering, requiring adaptation to heterogenous link prediction (LP). We adapt DBLP for heterogenous LP by splitting source nodes within DBLP based on author terms and target nodes from: paper, conference, and terms categories [2]. Train, validation, and testing splits are chosen randomly following 60/20/20. Negative edges are sampled randomly, following practices set within OGB [3]. As shown within Table 1, we demonstrate that FLEX improves mean AUC to 98.28, a significant result given our well-tuned GCN baseline with a mean AUC of 98.04.
> >
> > **shown below** (Table 1 -- AUC on the DBLP dataset, before (GCN) and after tuning with FLEX (GCN+FLEX.)
> > | GCN | GCN+FLEX |
> > |-----------|------|
> >  | 98.04 ± 0.20 | 98.28 ± 0.38 |
> >
> > The current DBLP result coupled with the pre-existing ogbl-collab results, indicate that FLEX can improve performance under complicated graph setting, even with well-tuned models. As such, we hope that these additional tests and revisions address all of reviewer NgWx's concerns. Thank you for attention and productive insights during this review process.
> >
> > [1] Fu, Xinyu, Jiani Zhang, Ziqiao Meng, and Irwin King. "Magnn: Metapath aggregated graph neural network for heterogeneous graph embedding." In Proceedings of the web conference 2020, pp. 2331-2341. 2020.
> > [2] https://pytorch-geometric.readthedocs.io/en/latest/_modules/torch_geometric/datasets/dblp.html#DBLP
> > [3] https://ogb.stanford.edu/docs/linkprop/

---

### Official Review · Reviewer_JXn7 · 2025-10-31

**Soundness:** 2
**Presentation:** 3
**Contribution:** 2
**Rating:** 4
**Confidence:** 4

**Summary:**

This paper introduces FLEX, a subgraph generation framework designed to enhance the generalization capability of graph neural networks in out-of-distribution link prediction tasks. Its core idea is to utilize a graph generation model trained collaboratively with the GNN to generate counterfactual subgraphs that differ structurally from training samples but share consistent node features, thereby enabling GNN fine-tuning. The method encourages structural diversity by maximizing the KL divergence between the posterior and prior distributions (with quadratic penalty), while preserving semantic relevance. The authors validate FLEX's effectiveness across multiple synthetic (LPShift) and real-world (ogbl-collab, Amazon Cross-Domain) OOD settings, conducting ablation studies, hyperparameter sensitivity analyses, and structural alignment evaluations.

**Strengths:**

1. OOD link prediction is a critical bottleneck for GNN deployment, yet existing work predominantly focuses on node/graph classification. This paper explicitly demonstrates that standard OOD generalization methods (e.g., IRM, CORAL) exhibit limited effectiveness in LP tasks (from Table 1), providing empirical evidence and establishing a robust problem motivation.


2. The paper employs k-hop subgraph generation instead of full-graph generation, introduces a labeling trick to enable GNNs to perceive target edges, and utilizes a semi-implicit VAE to balance generation quality and scalability.

3. Formally define meaningful structural differences through counterfactuals and feature condition equivalence, and prove in Appendix B that generated samples can scale training support sets to cover the test distribution from a set-theoretic perspective.

4. The experimental results showed improvement.

**Weaknesses:**

1. The paper repeatedly cites Pearl's causal framework, yet the FLEX generation process does not model structural equation models or interventions. Instead, it achieves structural differences solely through KL divergence maximization. This approach aligns more closely with diversity sampling in data augmentation than with counterfactuals in a strict causal sense.
2. Gamma significantly impacts performance, but selection relies on grid search. In real-world out-of-distribution scenarios where the test distribution is unseen, how can gamma be adaptively set. If gamma is set too high, resulting in overly sparse graphs, critical structural information may be lost.
3. FLEX has not been directly compared with recent OOD learning methods on LP tasks. While it is noted that these methods do not directly optimize LP generalization, experimental evidence is required to demonstrate that FLEX outperforms these generative OOD approaches.
4. Appendix B's Theorem 1 assumes that the generated sample set S satisfies  \mu(S\cap(U\T))>0. However, in practice, the Lebesgue measure of discrete graph spaces is zero. This assumption holds under continuous approximation but does not address discretization error.

**Questions:**

please see weakness.

---

> ### Author Response · Authors · 2025-11-24
> **Response to Reviewer JXn7 (1/2)**
>
> 1. "The paper repeatedly cites Pearl's causal framework, yet the FLEX generation process does not model structural equation models or interventions. Instead, it achieves structural differences solely through KL divergence maximization. This approach aligns more closely with diversity sampling in data augmentation than with counterfactuals in a strict causal sense."
>
> Thank you, the distinction for diversity sampling and strict causal intervention is important for quantifying FLEX's contribution.
> * We follow the convention of a counterfactual link in link prediction by considering the casual model in [1] (see Figure 2 within [1]). [1] derive that learned features across both $node_i$ ($n_i$) and $node_j$ ($n_j$) are applicable to a given treatment across both nodes ($T_{ij}$). CFLP applies the combination of ($n_i, n_j, T_{ij}$) to initiate causal intervention and learn on a counterfactual adjacency matrix ($A_{CF}$), promoting learning for a given predictor previously-trained on the untreated adjacency matrix ($A$).
> * FLEX parametrizes CFLP's approach by maximizing a learnable posterior conditioned by a structural embedding (treatment), approximating the causal intervention within CFLP's structural model to generate diverse training data. We conceptualize the newly-conditioned samples as link-counterfactual, since they **add structure not-previously captured within the original training data while preserving the original feature label**, as shown within our revised paper's Figure 7 and 8.
>
> 2. "Gamma significantly impacts performance, but selection relies on grid search. In real-world out-of-distribution scenarios where the test distribution is unseen, how can gamma be adaptively set. If gamma is set too high, resulting in overly sparse graphs, critical structural information may be lost."
>
> Thank you for the interesting question! We start with two of our key reasoning steps for applying grid-search and propose a promising alternative:
> * Effective hyperparameter selection for OOD generalization remains an open problem [1], we applied grid-search on $\gamma$ to avoid biased assumptions from target sample selection.
> * FLEX often benefits from the assumption that much of link-structured data during testing is sparser than training data (i.e. older papers have more time to accumulate citations). During our testing, higher gamma values were only problematic with the dense test samples from 'Forward' LPShift splits, as shown by the shift in $\gamma$ values in our revised paper's Table 9.
> * In order to overcome this effect, we apply a 'CN-Matching' algorithm on LPshift's CiteSeer 'Forward - CN' dataset. The algorithm dynamically adjusts gamma as follows: 1) Computes CN scores on training subgraphs, 2) Pulls training subgraphs with CN = 1 as validation samples, 3) adjust FLEX's gamma for each generated subgraph's structure to match valid subgraphs sample distribution (Similar to the Training Domain-Validation Set described in [1]). This rebuttal's Table 1 (shown below), shows 'CN-Matching' enabling **comparable performance improvements without the need for significant grid-search tuning** [2].
>
> **shown below** (Table 1 -- Training Structure-Validation Set. CN-Matching is applied to training samples to stabilize subgraph structure)
> | Sample Distribution | Original | 50% FLEX Subgraphs | 100% FLEX Subgraphs | FLEX w/ No 'CN-Matching' |
> |------|------|------|------|-----|
> | Hits@20 | 67.22 ± 2.30 | 68.38 ± 1.62 | 68.49 ± 1.41 | 68.24 ± 1.30 |
>
> [1] Gulrajani, Ishaan, and David Lopez-Paz. "In search of lost domain generalization." arXiv preprint arXiv:2007.01434 (2020).
> [2] Ben-David, Shai, et. al "A theory of learning from different domains." Machine learning 79, no. 1 (2010): 151-175.

---

> ### Author Response · Authors · 2025-11-24
> **Response to Reviewer JXn7 (2/2)**
>
> 3. "FLEX has not been directly compared with recent OOD learning methods on LP tasks. While it is noted that these methods do not directly optimize LP generalization, experimental evidence is required to demonstrate that FLEX outperforms these generative OOD approaches."
>
> * We thank the reviewer for acnkowledging how many recent OOD learning methods are not designed for LP tasks.
>     * 1) GOLD [3] is designed exclusively for detecting OOD graph samples. So, our investigation focuses on uplifting EERM [4] from graph/node classifcation to link prediction. We test EERM since it applies graph generation to improve OOD performance on graph data.
>     * 2) We include EERM results within this rebuttal's Table 1 and our revised paper's Table 3. In our evaluation, EERM goes OOM on every tested dataset besides Cora and CiteSeer. EERM also provides no performance improvement across Cora and CiteSeer.
>
> **shown below** (Table 2 -- Revised with EERM and original 'GCN', 'GCN+FLEX' results. Computers, Photo, PubMed, Collab, and PPA are not included since EERM (k=2) experiences OOM error when tested on a single A6000 GPU (same setting as FLEX).)
> | Dataset | Direction | Type | EERM | GCN | GCN+FLEX |
> |----|----------|------|-------|-------|-----------|
> |Cora| Forward | CN | 66.89 ± 2.78 | 67.18 ± 2.43 | 68.24 ± 1.30 |
> |CiteSeer| Forward | CN | 55.93 ± 1.49 | 56.22 ± 1.31 | 57.78 ± 0.08 |
> |Cora| Forward | PA | 68.41 ± 3.57| 68.88 ± 3.34 | 70.83 ± 0.41 |
> |CiteSeer| Forward | PA | 54.87 ± 5.03| 55.13 ± 5.30 | 56.58 ± 5.22 |
> |Cora| Forward | SP | 42.91 ± 2.09 | 44.60 ± 2.57 | 45.85 ± 0.24 |
> |CiteSeer| Forward | SP | 23.04 ± 3.28 | 24.82 ± 3.40 | 29.91 ± 0.19 |
> |Cora| Backward | CN | 39.85 ± 5.21 | 41.03 ± 5.68 | 43.96 ± 1.18 |
> |CiteSeer| Backward | CN | 38.39 ± 1.43 | 39.92 ± 1.09 | 44.87 ± 0.32 |
> |Cora| Backward | PA | 35.84 ± 2.10 | 37.67 ± 2.87 | 39.70 ± 0.26 |
> |CiteSeer| Backward | PA | 36.72 ± 1.89 | 38.00 ± 1.24 | 40.07 ± 0.14 |
> |Cora| Backward | SP | 11.74 ± 8.91 | 16.98 ± 5.12 | 22.09 ± 1.10 |
> |CiteSeer| Backward | SP | 23.94 ± 3.68 | 26.67 ± 3.49 | 28.25 ± 1.23 |
>
> [3] Wang, Danny, Ruihong Qiu, Guangdong Bai, and Zi Huang. "Gold: Graph out-of-distribution detection via implicit adversarial latent generation." arXiv preprint arXiv:2502.05780 (2025).
> [4] Wu, Qitian, Hengrui Zhang, Junchi Yan, and David Wipf. "Handling distribution shifts on graphs: An invariance perspective." arXiv preprint arXiv:2202.02466 (2022).
>
> 4. "Appendix B's Theorem 1 assumes that the generated sample set S satisfies \mu(S\cap(U\T))>0. However, in practice, the Lebesgue measure of discrete graph spaces is zero. This assumption holds under continuous approximation but does not address discretization error."
>
> * This is an excellent observation, we clarify that our encoding scheme $(Π:G→X)$ maps sets $(T,U,S)$ into a latent, continuous representation space $(X⊆R^2d)$; effectively enhancing resolution of potential coverage. We use $X$ interchangeably with an explicit latent space $Z$. Therefore, coverage for $(T,U,S)$ is determined in the embedding space ($Z$) and made learnable through the re-parametrization trick, before being sampled and decoded into discrete structural features ($X$). We include this as Assumption 2 within the revised Appendix B.
> *  We also note discretization error within neural systems is a fundamental problem worthy of research beyond the scope of this work [5].
>
> [5] Lanthaler, Samuel, Andrew M. Stuart, and Margaret Trautner. "Discretization error of Fourier neural operators." arXiv preprint arXiv:2405.02221 (2024).

---

### Official Review · Reviewer_EKCx · 2025-11-01

**Soundness:** 2
**Presentation:** 2
**Contribution:** 2
**Rating:** 4
**Confidence:** 3

**Summary:**

This paper proposes FLEX, a framework for improving out-of-distribution (OOD) link prediction in graphs. It co-trains a graph neural network with a semi-implicit variational autoencoder (SIG-VAE) that generates link-conditioned subgraphs—synthetic, counterfactual examples meant to expand the structural diversity of training data. By training on both real and generated subgraphs, the model aims to generalize better to unseen graph structures. Experiments on LPShift and real datasets are conducted to validate the proposed method.

**Strengths:**

1. The motivation is clear and intuitive.

2. The evaluation is conducted on four datasets with diverse shift schemes, and an ablation study is also provided.

**Weaknesses:**

1. I’m not fully convinced by the performance improvements shown in Tables 1 and 3. AUC scores below 0.5–0.6 are essentially trivial, indicating near-random predictions. Although FLEX often yields statistically significant gains, improvements such as 50% → 52% provide limited practical utility. The per-dataset breakdown in Table 3 further shows that GCN+FLEX frequently increases AUC while remaining in the trivial range. In some cases (e.g., Backward–PA), FLEX even degrades GCN’s AUC from 73% to 59%, effectively turning a non-trivial score into a trivial one. In addition, the “average gain” reported for NCN+FLEX is somewhat misleading—it should be compared against its backbone (NCN), not against GCN.

2. Table 5 reveals a substantial computational burden introduced by FLEX. The preprocessing and co-training steps are notably expensive, raising concerns about scalability to larger graphs or real-time applications. It is surprising that preprocessing even a small dataset like CiteSeer takes more than six hours.

3. It would strengthen the empirical validation to include results on a more diverse set of GNN backbones, such as GAT or GIN, to test the general applicability of FLEX.

4. It is not obvious how standard OOD generalization methods (e.g., IRM, VREx, GroupDRO) are adapted for link prediction tasks in this work. Providing implementation details or specific design choices for these adaptations would help the reader better assess fairness and reproducibility.

Minor issue:
All parentheses are printed incorrectly.

**Questions:**

N/A

---

> ### Author Response · Authors · 2025-11-24
> **Response to Reviewer EKCx (1/2)**
>
> 1. "I’m not fully convinced by the performance improvements shown in Tables 1 and 3. AUC scores below 0.5–0.6 are essentially trivial, indicating near-random predictions. Although FLEX often yields statistically significant gains, improvements such as 50% → 52% provide limited practical utility. The per-dataset breakdown in Table 3 further shows that GCN+FLEX frequently increases AUC while remaining in the trivial range. In some cases (e.g., Backward–PA), FLEX even degrades GCN’s AUC from 73% to 59%, effectively turning a non-trivial score into a trivial one. In addition, the “average gain” reported for NCN+FLEX is somewhat misleading—it should be compared against its backbone (NCN), not against GCN."
>
> Thank you for bringing this to our attention, we provide four points:
> * We measured performance with three separate metrics, this ensures consistency between our tests and previous works: 1) 'Forward' and 'Backward' = Hits@20 [1], 2) real-world ogbl-collab = Hits@50 [2], 3) Photos and Computers = AUC [3]. We have added a 'Metric' column to our revised Table 1 to better distinguish results.
> * The pre-existing 'Avg (Δ%)' was calculated from the raw model scores across all tested datasets, which was influenced by outliers. We have revised our submission's Table 1 to consider the average relative increase across the aggregated results, GCN+FLEX = '4.41%' and NCN+FLEX = '9.56%', respectively.
> * Hits@20 and Hits@50 are ranking metrics, which are sensitive to dataset imbalance from noisy samples. So, as shown in Table R1, FLEX reliably inducing a 5-10% relative performance increase indicates the effectiveness of our proposed method.
> * Additionally, Table 1 within our original submission demonstrates that FLEX raises the raw AUC scores for Photo->Computers from 87.48 to 91.16 and Computers->Photo from 83.87 to 91.36, **indicating 3.68% and 7.49% respective increases on non-trivial AUC scores**.
>
> **shown below** (Table R1 -- Additional results for baseline HL-GNN, GIN, GAT models and their FLEX counterparts.)
> | Direction | Type | HL-GNN | HL-GNN+FLEX | GAT | GAT+FLEX | GIN | GIN+FLEX |
> |-----------|------|--------|-------------|-----|----------|-----|----------|
> | Forward | CN | 36.16 ± 3.30 | 37.81 ± 2.71 | 41.28 ± 2.82 | 42.43 ± 2.34 | 41.43 ± 7.56 | 43.50 ± 5.97 |
> | Forward | PA | 49.21 ± 5.35 | 51.12 ± 2.86 | 51.15 ± 2.86 | 52.52 ± 2.19 | 51.26 ± 4.27 | 52.83 ± 2.89 |
> | Forward | SP | 42.24 ± 2.60 | 43.47 ± 1.62 | 33.27 ± 5.58 | 35.99 ± 3.01 | 32.48 ± 3.93 | 34.56 ± 2.68 |
> | Backward | CN | 22.59 ± 3.92 | 24.24 ± 2.71 | 28.93 ± 2.42 | 30.10 ± 1.33 | 22.42 ± 4.27 | 23.97 ± 2.78 |
> | Backward | PA | 16.12 ± 4.85 | 17.82 ± 3.90 | 43.84 ± 4.23 | 45.30 ± 2.73 | 23.93 ± 10.01 | 25.24 ± 2.36 |
> | Backward | SP | 20.92 ± 4.08 | 22.39 ± 3.68 | 13.89 ± 3.61 | 17.07 ± 0.84 | 16.82 ± 6.14 | 18.70 ± 2.09 |
> | -- | **Δ%** | -- | **+6.04%** | -- | **+7.34%** | -- | **+6.35%** |
>
> [1] Li, Juanhui, et. al "Evaluating graph neural networks for link prediction: Current pitfalls and new benchmarking." Advances in Neural Information Processing Systems 36 (2023): 3853-3866.
> [2] Hu, Weihua, et. al. "Open graph benchmark: Datasets for machine learning on graphs." Advances in neural information processing systems 33 (2020): 22118-22133.
> [3] Dong, Yuxiao, et. al. "Link prediction and recommendation across heterogeneous social networks." In 2012 IEEE 12th International conference on data mining, pp. 181-190. IEEE, 2012.
>
> 2. "Table 5 reveals a substantial computational burden introduced by FLEX. The preprocessing and co-training steps are notably expensive, raising concerns about scalability to larger graphs or real-time applications. It is surprising that preprocessing even a small dataset like CiteSeer takes more than six hours."
>
> * We'd like to emphasize that Table 5 demonstrates **CFLP's pre-processing time** to derive non-parametric counter-factuals, **not FLEX**.
> * Table 6 compares the per-epoch training efficiency of FLEX (top-row) to CFLP (bottom-row) on the GCN baseline, where CFLP takes 20x longer than FLEX on PubMed. On the larger ogbl-collab and ogbl-ppa datasets--CFLP incurs an OOM error, whereas FLEX successfully scales to both datasets on a single A6000 GPU.
> * Table 7 compares the inference speed of the baseline GCN to FLEX, where **FLEX imparts no slow-down to GCN**, largely since FLEX's generative step operates in-sequence before GCN's inference step.

---

> ### Author Response · Authors · 2025-11-24
> **Response to Reviewer EKCx (2/2)**
>
> 3. "It would strengthen the empirical validation to include results on a more diverse set of GNN backbones, such as GAT or GIN, to test the general applicability of FLEX."
>
> Thank you for the suggestion, we include additional tests with HL-GNN, GAT, and GIN for Table R1 and a revised Table 3 within our submission. Both tables empirically validate that FLEX **consistently improves the performance of each backbone model**.
>
> 4. "It is not obvious how standard OOD generalization methods (e.g., IRM, VREx, GroupDRO) are adapted for link prediction tasks in this work. Providing implementation details or specific design choices for these adaptations would help the reader better assess fairness and reproducibility."
>
> We agree that further details for OOD generalization is important to ensure fairness and reproducibility, we extend the original descriptions within Appendix G as follows:
>
> * We integrate: IRM, VREx, GroupDRO, DANN, CORAL, and EERM from GOOD [4] into a baseline GCN with MLP link-predictor.
>
> * All tested methods operate under the same assumption as FLEX, that the testing distribution remains unseen until evaluation. The environmental subsets used to test OOD generalization methods were randomly-sampled and then grid-searched across equal-sized subsets, e = {3, 4, 5} to resolve the final e = 3 [5].
>
> [4] Gui, Shurui, et. al. "Good: A graph out-of-distribution benchmark." Advances in Neural Information Processing Systems 35 (2022): 2059-2073.
> [5] Gulrajani, Ishaan, and David Lopez-Paz. "In search of lost domain generalization." arXiv preprint arXiv:2007.01434 (2020).
>
> 6. "Minor issue: All parentheses are printed incorrectly"
> * Our revised paper has updated this problem, thank you again for your attentive review!

---

### Author Response · Authors · 2025-12-03
**Rebuttal Summary -- Thank you to Reviewers and Chairs**

We thank the reviewers and chairs for a fruitful rebuttal, their efforts have substantially-enhanced our work. We now summarize how all reviewer concerns have been addressed:

**1. Experimental breadth and clarity**

* We clarified metrics (Hits@20/50 vs AUC) via a new *Metric* column in Table 1 and corrected “Avg (Δ%)” to average **relative** gains per dataset: +4.41% for GCN+FLEX, +9.56% for NCN+FLEX, and +6–7% for HL-GNN/GAT/GIN (Table R1).
* FLEX improves *non-trivial* scores, e.g., Amazon Photos→Computers AUC 87.48→91.16 and Computers→Photos 83.87→91.36, showing practically meaningful uplift rather than marginal gains near random performance.
* We expanded backbone coverage to **HL-GNN, GAT, and GIN**, where FLEX consistently yields ~5–8% relative improvements across LPShift splits (Table R1).
* We added a **heterogeneous DBLP for link prediction** (adapting DBLP from node classification): FLEX improves a strong GCN baseline from 98.04±0.20 to 98.28±0.38 AUC, indicating applicability to more complex graphs.
*  We add **EERM** tests, showing the method goes OOM on larger datasets or fails to improve Cora/CiteSeer, while FLEX consistently helps, underscoring FLEX’s empirical advantage.
* We also fixed typographical issues to improve overall presentation.

**2. Framework scalability and efficiency**

* Very large preprocessing times in Table 5 belong to **CFLP**, not FLEX.
* Tables 5–8 and Appendix I demonstrates runtime and memory comparisons: FLEX is up to **20× faster than CFLP** per epoch on PubMed and does **not slow down inference**. FLEX successfully runs on dense ogbl-ppa where CFLP hits OOM.
* Appendix F provides a **formal time and memory complexity analysis**, decomposing encoder, sampler, decoder, and backbone GNN, and explaining why hash-based methods like BUDDY are prohibitively expensive in a *generative* loop, whereas FLEX remains scalable at the subgraph level.

**3. Causual and risk-based theory**

* Appendix B now includes **Proposition B.1**, **Corollary B.3**, **Remark 4**, and **Assumption 2**, which together show:

  * risk can be bounded in terms of the remaining test mass outside the augmented training support;
  * successful FLEX tuning enlarges train–test overlap in representation space and therefore reduces estimated OOD risk;
  * the **KL-based structural diversity objective** directly increases this overlap in latent space.
  * graphs are mapped into a continuous latent space, where Lebesgue measures can be more reasonably estimated before decoding to discrete structure.
* For causality: FLEX is now framed as a **parametric link-counterfactual generator** that *approximates* CFLP-style interventions via a structurally conditioned posterior, rather than a full SEM, aligning the paper more honestly with causal and data-augmentation viewpoints.

**4. Hyperparamater ablations and deep implementation testing**

* Appendix J (Figures 10–11) now provides a thorough **ablation/sensitivity study**:

  * varying the ratio of FLEX-generated subgraphs, noise in semi-implicit sampling parameters (K, J);
  * showing that **joint** training of FLEX+GNN is necessary (training a new GNN from scratch on FLEX subgraphs quickly collapses performance);
  * demonstrating that overly large KL weights produce noisy subgraphs that hurt performance.
* We introduce **CN-Matching** as a label-free strategy to adapt γ in practice: it aligns generated subgraph CN statistics with a validation subset and stabilizes performance on challenging LPShift “Forward–CN” splits without heavy grid search.
* We include **visualizations of generated subgraphs** in Figures 7–8 to complement Appendix H's structural statistics, confirming both structural diversity and semantic consistency relative to train/test subgraphs.
* Appendix K (Figure 12) now details the **SIG-VAE architecture**, Appendix G (Table 9) lists **loss coefficients and hyperparameters**, and we clarify the **sampling and selection strategy** from the semi-implicit posterior. A reference implementation is provided in the supplementary zip.
* Appendix G includes details on **IRM, VREx, GroupDRO, DANN, CORAL, and EERM** are applied in a  common GCN+MLP LP backbone.

---

### Meta-Review · Area_Chair_9xeb · 2026-01-12

**Summary:**

The paper introduces FLEX, an approach for improving OOD link prediction. The key idea is to use a generative model (semi-implicit VAE) to generate structurally diverse subgraphs. The paper frames these as "counterfactual" subgraphs, however, as Reviewer JXn7 notes, these are better viewed as augmentations which are encouraged to be diverse. In the rebuttal the authors claim that "FLEX is now framed as a parametric link-counterfactual generator that approximates CFLP-style interventions via a structurally conditioned posterior, rather than a full SEM, aligning the paper more honestly with causal and data-augmentation viewpoints.". However, based on the changes indicated in blue in the revised manuscript, this update framing is not clearly reflected in the text.

Reviewer JXn7 also had concerns about the selection of gamma. The rebuttal for why grid-search is necessary is not fully convincing. The sensitivity to gamma is indeed a valid limitation of the method which can be a show stopper in practice. In the rebuttal, the authors propose a new ad-hoc algorithm to dynamically adjusts gamma. This addition is appreciated, but the authors only show results for one dataset (which I also believe is the synthetic one).

The question about assumptions behind Theorem 1 is not fully answered in the rebuttal, but rather side-stepped. In any case the issue of discretisation error needs to be discussed in the paper, even though this may be a fundamental orthogonal problem.

Reviewer EKCx raised concerns about the performance improvements. Those concerns were valid, assuming the metric is AUC in all raws, but the authors clarified in the rebuttal which rows correspond to the HITS@20 metric and which to AUC. Nonetheless, the authors state "FLEX improves a strong GCN baseline from 98.04±0.20 to 98.28±0.38 AUC, indicating applicability to more complex graphs." However, it is not clear whether the error bars indicate standard error, standard deviation, or a confidence interval,  which would have completely different implications for the statistical significance of the improvement. The improvement in some of the other metrics seems more substantial. Still, it's not clear why all synthetic datasets are evaluated using Hits@20, while the "real" ogbl-collab is evaluated with Hits@50. What is the rationale for this choice? How does the performance for ogbl-collab change for Hits@K when K is varied? In general, I think the paper would benefit from more extensive evaluation of the method on real-world datasets, since it's not clear how the insights from the synthetic datasets transfer.

Reviewer NgWx raised several issues which the authors attempted to address in the rebuttal. However, the reviewer claims that "many issues—such as the theoretical analysis, ablation studies, and complex-graph evaluations—remain unresolved". I agree, especially regarding the complex-graph evaluations.  The reviewer also mentioned that "several revisions mentioned in the response (e.g., Proposition B.1, Corollary B.3, Remark 4, Figure 12) do not appear in the updated manuscript." however, the authors clarified that they forgot to upload the updated manuscript. Looking at the changes in blue, it seems that some of these issues have been addressed, specifically the ablation studies.

In general, while the theoretical contributions are interesting, to me it remains unclear whether they specifically explain why FLEX works better since it seems the analysis can be applied to any data augmentation method. Relatedly, comparing with augmentation methods seems prudent to better understand the benefits of FLEX.

Reviewer Xvsx was concerned about limited backbones, and the authors added GAT, GIN and HL-GNN, however only on synthetic datasets. Their other main concern about efficiency and other encodings were sufficiently addressed. However, I agree with the reviewer that results on OGBL-Citation2 should be added (despite the edge density comment). If, as the authors claim, "SOTA graph generative models rarely scale beyond 5000 nodes" indeed holds, than this is an important limitation that should be discussed in the paper. Even though FLEX uses subgraphs, it's not clear if the improvements will hold graphs with significantly more nodes.

Overall, I think the proposed approach has merit, however the authors need to address the aforementioned issues before the paper is ready for publication. In the revision, I suggest the authors to focus on clearly delineating their approach from data augmentation (empirically and theoretically), avoid overstating causal claims, and clearly acknowledge all the limitations, and specifically gamma. Most importantly, the real-world evaluation needs to be expanded.

**Reviewer Concerns:**

The authors addressed many of the concerns including: clarified metrics, additional backbones, heterogeneous DBLP experiment, comparisong to EERM, clarified preprocessing time and added complexity analysis, ablation studies.

The key remaining concerns are: regarding gamma sensitivity (despite the added CN-Mathcing), the casual framing and the significance of the theoretical analysis, the interpretation of the results, the lack of larger and more complex real-world datasets.

**Reviewer Scores:**

Reviewer NgWx would have likely maintained their score, even after the seeing the new updated manuscript which the authors forgot to upload.

There is some chance that Reviewer Xvsx and Reviewer EKCx could have increased their score.

---

### Decision · Program_Chairs · 2026-01-26

Reject